# Flexible Deep Learning Techniques for Cox Models with Data Integration

## Abstract

Prognostic prediction using survival analysis faces challenges due to complex relationships between risk factors and time-to-event outcomes. Deep learning methods have shown promise in overcoming these challenges, but their effectiveness is often dependent on large datasets. Deep learning methods implemented on moderate or small datasets often suffer from severe problems, such as insufficient training data, overfitting, and difficulty in tuning hyperparameters. To address these issues and enhance the accuracy of prognosis predictions, this paper introduces a novel, flexible deep learning framework for integrating information from external risk models with internal time-to-event data. Our approach accommodates both homogeneous and heterogeneous settings, allows for the integration of risk scores derived from various external models, and leverages the power of deep neural networks to capture complex non-linear relationships. The approach makes use of a generalized Kullback-Leibler distance measure between survival distributions, which is included as a penalization term in the estimation procedure. We demonstrate the improved predictive accuracy of our approach through extensive simulations and real-world applications.

## 1 Introduction

Prognosis prediction is an important topic in survival analysis. Historically, research aimed at predicting survival outcomes has largely been confined to individual datasets. These datasets often have limitations, such as rare event rates, small sample sizes, high dimensionality, and low signal-to-noise ratios. To overcome these limitations, transfer learning in survival analysis has been proposed to improve prediction accuracy by transferring knowledge from external, pre-trained models into the analysis of newly collected data (Huang et al., 2016; Chen et al., 2021). However, traditional integrated approaches, such as the integrated Cox proportional hazards model, often face limitations in prognostic prediction capabilities due to their dependence on the linearity and proportional hazards assumptions. In reality, the relationship between event times and risk factors can be intricate, often involving non-linear effects, influences that vary over time, and interactions. To effectively capture the complexities of integrated time-to-event data, it is essential to employ computationally efficient deep learning techniques.

Deep learning has been increasingly applied to survival analysis in recent years. Katzman et al. (2018) extend the classical Cox proportional hazards model by parameterizing the risk function using neural networks. The resulting algorithm (termed DeepSurv) models the risk function as a non-linear transformation of input covariates, allowing the capture of complex nonlinear effects and interactions. Cox-Time (Kvamme et al., 2019) further extends the Cox model by removing the proportionality constraint and allowing the risk function to depend on time. This is achieved by parameterizing the risk function using a neural network that takes both covariates and time as input. Discrete-time methods, including DeepHit (Lee et al., 2018), Nnet-survival (Gensheimer & Narasimhan, 2019), DRSA (Ren et al., 2019), and SSMTL (Chi et al., 2021), consider time as discrete and employ classification techniques with binary event indicators for each time interval. Some methods, such as DeepHit, SSMTL, and SSCNN (Agarwal et al., 2021) also incorporate ranking-based losses to modify model performance. These diverse methodological advances demonstrate the potential of deep learning in enhancing survival analysis by providing more flexible modeling options, the ability to learn from complex and high-dimensional data, and improved predictive performance compared to traditional statistical methods.

Although deep learning methods have shown significant promise, their effectiveness often relies on large datasets to fully capture the complex patterns and relationships. However, in many real-world scenarios, sample sizes are often limited. This challenge is further exacerbated in time-to-event data, where high censoring rates frequently result in low event rates. In such situations, deep learning methods often encounter severe problems including insufficient training data, model instability, overfitting, and difficulty in tuning hyperparameters. This "small data" challenge is a primary motivation for transfer learning, where knowledge from a large source domain can be used to regularize and guide a model being trained on a smaller target domain.

This research is motivated by The Michigan Urological Surgery Improvement Collaborative (MU-SIC) study (Womble et al., 2014), which provides individual-level clinical and genomic data for 378 men with prostate cancer who were treated with surgery. The outcome of interest is time to biochemical recurrence. Given the small sample size, integrating external information from validated prognostic tools presents a promising avenue to enhance survival prediction. One example of such external information is the STAR-CAP (Staging Collaboration for Cancer of the Prostate) system, which is a point-based clinical prognostic tool for localized prostate cancer (Dess et al., 2020) assigning weights to various clinical factors to categorize patients into distinct risk groups. This system provides a standardized and globally applicable framework for risk assessment. However, the points-based nature of STAR-CAP pose challenges for integration with existing methods that rely on survival probabilities or risk scores.

Previous research has explored the knowledge transfer from external information at the summary level with internal survival data. Huang et al. (2016) developed an empirical likelihood approach under a strong homogeneity assumption that internal data are collected from the same underlying probability distribution from which external summary-level information is derived. To relax the homogeneity assumption, Chen et al. (2021) proposed an adaptive estimator that penalizes the potential discrepancy between data sources, a challenge known in machine learning as domain shift. However, this method only applies to external subgroup survival rates at certain time points defined based on a few categorical variables. Alternatively, Kullback-Leibler (KL) information (Kullback & Leibler, 1951) has been applied to data integration. Specifically, Liu & Shum (2003), Schapire et al. (2005) and Jiang et al. (2016) developed KL-based data integration techniques for binary and ordinal outcomes. In the context of survival analysis, Wang et al. (2025) proposed a discrete failure time modeling procedure using discrete hazard-based KL information as a metric to quantify discrepancies between published models and internal datasets, demonstrating improved prediction performance in a kidney transplant study. Recently, Wang et al. (2023) developed a partial likelihood-based KL approach for continuous survival data, integrating external risk scores with internal Cox models while addressing population heterogeneity, or domain shift.

Although the data integration methods mentioned above have proven effective, they are primarily designed to integrate external survival information that are linked to probabilities derived from traditional statistical models, such as the Cox proportional hazards model. However, these methods are not directly applicable to the STAR-CAP system. In STAR-CAP, patients are classified into distinct risk groups based on a weighted sum of clinical factors. Such groups, which may not provide direct probability interpretation, are used to classify patients rather than predict survival probabilities. These issues preclude the application of the aforementioned integration methods that rely on a probability-based metric to measure the discrepancy between the internal data and historical models. Furthermore, existing integration procedures are based on linear relationships between predictors and survival outcomes. It is computationally challenging to extend these methods by allowing for a flexible, non-linear function to model the relationship between covariates and the log-hazard function.

To robustly incorporate external ranking knowledge and improve discriminatory power, we propose a generalized KL-based transfer learning framework (named NNCoxKL). The proposed integration framework is flexible in the following key aspects: (1) it leverages the power of deep neural networks to capture complex non-linear relationships and interactions to gain improved predictive accuracy; (2) it is robust to the domain shift between the source of external knowledge and the target data, automatically integrating diverse information sources optimally; and (3) it flexibly transfers knowledge from the external information whether it is a score produced by a survival model, or rankings based on clinical knowledge. (4) it only assumes that, for each subject in the internal dataset, we can compute an external risk score using a subset (or transformation) of the internal covariates. The remaining parts of this paper are organized as follows: Section 2 introduces the transfer learning

framework. Section 3 presents a thorough evaluation through simulation studies, with additional results from real-world data applications provided in the Appendix. Section 4 concludes with a discussion.

## 2 METHODS

### 2.1 INTERNAL FLEXIBLE PROPORTIONAL HAZARDS MODEL

We first introduce the proportional hazards model for the target internal cohort, which consists of individual-level data with time-to-event outcomes and risk characteristics.

Let $T_i$ denote the event time of interest and $C_i$ be the censoring time for the $i$-th individual, where $i = 1, \ldots, n$, and $n$ is the sample size of the internal cohort. Let $\mathbf{Z}_i$ denote a $p$-dimensional covariate vector for each subject. We assume that $T_i$ is independently censored by $C_i$, given $\mathbf{Z}_i$. The observed time is $X_i = \min(T_i, C_i)$, and the event indicator is $\delta_i = I(T_i \leq C_i)$. We consider proportional hazards models, which are specified by the following hazard function:

$$\lambda(t; \mathbf{Z}_i) = \lim_{dt \to 0} \frac{1}{dt} Pr(t \leq T_i < t + dt | T_i \geq t, \mathbf{Z}_i) = \lambda_0(t) \exp\{r(\mathbf{Z}_i, \boldsymbol{\beta})\},$$

where $\lambda_0(t)$ is an arbitrarily unspecified baseline hazard function, $r(\mathbf{Z}_i, \boldsymbol{\beta})$ specifies the log-relative risk relationship between the covariates $\mathbf{Z}_i$ and the hazard function, and $\boldsymbol{\beta}$ are the regression parameters. Throughout, we fix the functional form of $r(\cdot, \cdot)$ in advance (e.g., linear Cox vs. a specified neural-network architecture), and use $\boldsymbol{\beta}$ to denote the full collection of trainable parameters of the chosen model. The corresponding log-partial likelihood is given by

$$\ell(\boldsymbol{\beta}) = \sum_{i=1}^{n} \delta_i \left[ r(\mathbf{Z}_i, \boldsymbol{\beta}) - \log \left\{ \sum_{j=1}^{n} Y_j(X_i) \exp\{r(\mathbf{Z}_j, \boldsymbol{\beta})\} \right\} \right], \tag{1}$$

where $Y_j(X_i) = I(X_j \geq X_i)$ is the at-risk indicator.

The classic Cox proportional hazards model is based on a linear function $r(\mathbf{Z}_i, \boldsymbol{\beta}) = \mathbf{Z}_i^\top \boldsymbol{\beta}$ (Cox, 1972). While the linearity assumption is often made, the Cox proportional hazards model may suffer when the true covariate-hazard relationship is non-linear and includes interactions. To allow more complex, potentially non-linear dependencies between covariates $\mathbf{Z}_i$ and the hazard function, we propose using a neural network for $r(\mathbf{Z}_i, \boldsymbol{\beta})$.

### 2.2 EXTERNAL RISK SCORES

Access to individual-level data from external studies is usually limited. Often only summary-level information can be obtained for the external model, for example from a publication. Specifically, we consider the situation that some external risk scores, $\tilde{r}(\mathbf{Z}_i)$, can be calculated based on external studies, where $\mathbf{Z}_i$ is the input covariate vector for the $i$th subject in the internal cohort.

In our proposed NNCoxKL framework, the form of the external risk scores $\tilde{r}(\mathbf{Z}_i)$ can be very flexible. They may originate from various models, such as neural networks, Cox proportional hazards models, boosting methods, or regularized techniques like Lasso. They can even be in the form of risk group classifications, such as those provided by the STAR-CAP system.

Furthermore, the covariates $\mathbf{Z}_i$ used in the external risk score may not perfectly match those of the internal cohort. The external model might have been developed using a subset of the internal covariates and outcome measures, which could be related to, but not identical to, the actual outcomes (e.g., a surrogate endpoint). This flexibility allows for the integration of diverse external information into our analysis, enhancing its applicability in real-world scenarios where data availability and consistency can be challenging.

### 2.3 GENERALIZED KL BASED TRANSFER LEARNING PROCEDURE

A unique challenge in transferring knowledge from external risk scores, such as STAR-CAP, is that these scores are derived for the purpose of patient classification rather than prediction of survival

probabilities. Often, these scores cannot provide any probability interpretation. These issues preclude the application of existing transfer learning methods.

To address this issue, we propose a solution motivated by the Bregman divergence (Bregman, 1967). For two vectors $\mathbf{p}$ and $\mathbf{q}$ and a strictly convex and differentiable function $G$, the Bregman divergence is defined to be the difference between the value of $G$ at point $\mathbf{p}$ and the value of the first-order Taylor expansion of $G$ around point $\mathbf{q}$ evaluated at point $\mathbf{p}$:

$$\mathrm{B}_G(\mathbf{p}\|\mathbf{q}) = G(\mathbf{p}) - G(\mathbf{q}) - \langle \triangledown G(\mathbf{q}), \mathbf{p} - \mathbf{q}\rangle,$$

where $\langle \cdot, \cdot \rangle$ means the inner product and $\triangledown G$ is the gradient vector.

To extract information from external risk scores, we cast the censored time-to-event data as a ranking problem. Specifically, assume that the internal cohort has $K$ unique failure times $t_1 < \ldots < t_K$. Let $R_k$ be the set of items at risk of failure at time $t_k^-$, just prior to time $t_k$. Thus, $R_k$ consists of all individuals who have not failed and are still under observation (uncensored) just before time $t_k$. For individual $i$ belonging to $R_k$, define a Plackett-Luce (Plackett, 1975) type of ranking metric,

$$\mathbf{p}_k(i) = \frac{\exp\{\tilde{r}(\mathbf{Z}_i)\}}{\sum_{j\in R_k}\exp\{\tilde{r}(\mathbf{Z}_j)\}},$$

where $\tilde{r}(\mathbf{Z}_i)$ is the external risk score, and the exponential transformation of the risk scores ensures that $\mathbf{p}_k(i)$ is non-negative. Similarly, define

$$\mathbf{q}_k(i) = \frac{\exp\{r(\mathbf{Z}_i,\boldsymbol{\beta})\}}{\sum_{j\in R_k}\exp\{r(\mathbf{Z}_j,\boldsymbol{\beta})\}}.$$

To measure the disparity between $\mathbf{p}_k$ and $\mathbf{q}_k$, consider the generalized KL divergence,

$$\mathrm{d}(\mathbf{p}_k\|\mathbf{q}_k) = \sum_{i\in R_k}\mathbf{p}_k(i)\log\frac{\mathbf{p}_k(i)}{\mathbf{q}_k(i)} - \sum_{i\in R_k}\mathbf{p}_k(i) + \sum_{i\in R_k}\mathbf{q}_k(i),$$

which is a special case of the Bregman divergence generated by the negative entropy function, $G(\mathbf{p}_k) = \sum_{i\in R_k}\mathbf{p}_k(i)\log\mathbf{p}_k(i)$. Note that both $\mathbf{p}_k$ and $\mathbf{q}_k$ are standard simplex; that is, $\sum_{i\in R_k}\mathbf{p}_k = \sum_{i\in R_k}\mathbf{q}_k = 1$. Thus, the generalized KL divergence $\mathrm{d}(\mathbf{p}_k\|\mathbf{q}_k)$ reduces to the usual KL divergence,

$$\mathrm{d}(\mathbf{p}_k\|\mathbf{q}_k) = \sum_{i\in R_k}\mathbf{p}_k(i)\log\frac{\mathbf{p}_k(i)}{\mathbf{q}_k(i)}.$$

To capture information across failure times $t_1,\ldots,t_K$, the accumulated generalized KL is then defined as

$$\mathrm{D}(\tilde{\mathbf{r}} \| \mathbf{r}) = \sum_{k=1}^{K}\mathrm{d}(\mathbf{p}_k\|\mathbf{q}_k), \tag{2}$$

which measures the disparity between the historical risk scores, $\tilde{\mathbf{r}} = \{\tilde{r}(\mathbf{Z}_i), i=1,\ldots,n\}$, and the internal risk scores, $\mathbf{r} = \{r(\mathbf{Z}_i,\boldsymbol{\beta}), i=1,\ldots,n\}$. To integrate information from both cohorts while allowing for the potential disparities, we combine the log-partial likelihood from the internal data and the accumulated generalized KL by constructing a penalized log-partial likelihood

$$\ell_\eta(\boldsymbol{\beta}) = \ell(\boldsymbol{\beta}) - \eta\,\mathrm{D}(\tilde{\mathbf{r}} \| \mathbf{r}), \tag{3}$$

where $\eta$ is a tuning parameter weighing the relative importance of external and internal risk scores. In the special case of $\eta = 0$, the penalized log-partial likelihood $\ell_\eta(\boldsymbol{\beta})$ is reduced to the log-partial likelihood of the internal model $\ell(\boldsymbol{\beta})$. Finally, the tuning parameter $\eta$ can be selected via $V\&VH$ cross-validation (Verweij & Van Houwelingen, 1993).

*Remark 1*: Wang et al. (2023) proposed a partial likelihood-based KL to transfer knowledge from external risk scores generated from Cox proportional hazards models. To incorporate more general external information, such as ranking-based risk scores, we employ the generalized KL divergence. This allows us to measure the disparities even when external risk scores are not derived from a proportional hazards model.

*Remark 2 (Temperature-Scaled External Risk).* We note that the divergence measure is not invariant to the rescaling of the external scores $\tilde{r}$. To address this in a principled manner, we

draw upon the Knowledge Distillation literature (Hinton et al., 2015; Guo et al., 2017) and introduce a temperature scaling parameter $\alpha > 0$. We define the external Plackett–Luce probabilities as: $\mathbf{p}_k^{(\alpha)}(i) = \frac{\exp\{\tilde{r}(\mathbf{Z}_i)/\alpha\}}{\sum_{j \in R_k} \exp\{\tilde{r}(\mathbf{Z}_j)/\alpha\}}$, where $\alpha$ acts as the temperature. As established in prior work (Hinton et al., 2015), this parameter controls the entropy of the target distribution: a lower $\alpha$ produces a sharper distribution (emphasizing high-risk patients), while a higher $\alpha$ softens the distribution. This formulation induces a family of divergences $D_\alpha(\tilde{\mathbf{r}} \| \mathbf{r})$ and a penalized objective $\ell_{\eta,\alpha}(\beta) = \ell(\beta) - \eta D_\alpha(\tilde{\mathbf{r}} \| \mathbf{r})$. Instead of fixing this parameter, we treat the pair $(\eta, \alpha)$ as hyperparameters to be tuned jointly (e.g., via Optuna), allowing the model to adaptively determine the optimal scaling for the external signal and improving robustness across domains.

## 2.4 DEEP NEURAL NETWORK

**Proposition 1** *Ignoring terms not involving $\beta$, the penalized log-partial likelihood is*

$$\ell_\eta(\beta) = \sum_{k=1}^{K} \sum_{i=1}^{n} \left[ \delta_i(t_k) + \frac{\eta Y_i(t_k) \exp\{\tilde{r}(\mathbf{Z}_j)\}}{\sum_{j=1}^{n} Y_j(t_k) \exp\{\tilde{r}(\mathbf{Z}_j)\}} \right] \left[ r(\mathbf{Z}_i, \beta) - \log \left\{ \sum_{j=1}^{n} Y_j(t_k) \exp\{r(\mathbf{Z}_j, \beta)\} \right\} \right] \tag{4}$$

$$\propto \sum_{i=1}^{n} \left[ \frac{\delta_i + \eta \tilde{\delta}_i}{1 + \eta} r(\mathbf{Z}_i, \beta) - \delta_i \log \left\{ \sum_{j=1}^{n} Y_j(t_i) \exp\{r(\mathbf{Z}_j, \beta)\} \right\} \right], \tag{5}$$

*where $\delta_i(t_k) = I(T_i \leq C_i, T_i = t_k)$ is the event indicator for subject $i$ at time $t_k$, and*

$$\tilde{\delta}_i = \sum_{k=1}^{K} \frac{Y_i(t_k) \exp\{\tilde{r}(\mathbf{Z}_i)\}}{\sum_{j=1}^{n} Y_j(t_k) \exp\{\tilde{r}(\mathbf{Z}_j)\}}$$

*is a predicted event indicator, defined using external risk scores.*

When comparing equations equation 1 and equation 4, a key aspect of our approach, demonstrated in Proposition 1, is that the modified objective function retains a structure similar to the traditional log-partial likelihood. Therefore, standard deep learning techniques can be easily implemented with the proposed procedure to allow complex architectures and maintain computational efficiency.

Specifically, to implement the proposed transfer learning based framework, we adopt a standard feed-forward architecture designed to model complex relationships between covariates and the hazard function. This network takes as input both the predictor variables and the external risk score information $\tilde{r}(\mathbf{Z}_i)$, and produces a single output node that represents the risk score $r(\mathbf{Z}_i, \beta)$. To enhance model performance, we standardize the input features and utilize Kaiming's uniform initialization (He et al., 2015) for weight initialization. To prevent overfitting, the loss function includes an L2 regularization term applied to the weights. The hidden layers employ rectified linear unit (ReLU) activation functions (Nair & Hinton, 2010). For optimization, we adopt the AdamW optimizer with weight decay and employ a learning rate scheduler (Loshchilov & Hutter, 2017). Early stopping techniques are also implemented (Prechelt, 2002). Hyperparameter optimization, including learning rate, weight decay coefficient, dropout rate, and number of hidden layers, is conducted using a random search approach. Implementation details on mini-batch optimization and its comparison with full-batch training are provided in Appendix A.10.

## 2.5 CONVEX-ANALYTIC INTERPRETATION.

To evaluate the proposed method computationally, we examine its numerical optimality through convex analysis. Specifically, Proposition 2 below demonstrates the numerical property of the proposed procedure. The penalized log-likelihood, $\ell_\eta$, is shown to minimize the convex combination of the relative entropy between the working model and two extremes: one based solely on the target cohort and the other based solely on the external model.

**Proposition 2** Let $P_n$ be the saturated model (Simon et al., 2011) on the target cohort. For a given $\eta \geq 0$, the model $P$ minimizing $(1 - \omega)D(P_n \| P) + \omega D(\tilde{P} \| P)$ is the proposed integrated model, where $\omega = \eta/(1 + \eta)$.

Minimizing the convex combination over all models, $P$, seeks a distribution close to both the empirical saturated model $P_n$ and the external distribution $\tilde{P}$, weighted by the tuning parameter. Since KL divergence is convex in its second argument (i.e. in $P$) and a weighted sum of convex functions is also convex, the integrated objective is convex in the space of distributions. Therefore, an optimal distribution minimizing the objective exists and is unique.

## 2.6 Model Complexity, Overfitting and the Role of Transfer Learning

A key consideration in neural network design is the trade-off between model complexity and the risk of overfitting. The number and configuration of hidden layers play a critical role in maintaining this balance. Deeper networks with more hidden layers have a higher capacity to model complex relationships within the data. However, they are also more susceptible to overfitting, especially when training data is limited. As demonstrated in Section 4.1, transfer learning can enrich the model's learning process, allowing it to effectively utilize the increased complexity of deeper networks. By transferring knowledge, the proposed procedure reduces the model's reliance on potentially limited patterns in the internal training data, thereby improving predictive performance while preserving generalizability.

The choice of stopping criteria is crucial in deep learning to prevent overfitting, where the model perfor ms well on training data but poorly on unseen data. Early stopping is a common technique that involves monitoring the performance of the model in a validation set during training and stopping the training process when the validation loss starts to increase (Prechelt, 2002). This approach helps ensure the selection of a model that generalizes well to new data. However, the choice of stopping criteria and the optimal stopping point can be sensitive, particularly when dealing with small sample sizes. Variations in performance introduced by different stopping criteria can have a significant impact on the final results.

By transfer learning from external information, our proposed NNCoxKL method reduces reliance on limited or unique patterns present only in the internal data. As a result, the NNCoxKL model's learning process becomes less sensitive to specific early stopping points, thereby mitigating the risk of overfitting and delivering more robust and improved performance.

## 3 Simulation Study

To assess the performance of the proposed NNCoxKL method, we conducted a simulation study evaluating NNCoxKL's prediction capabilities under various scenarios, including its integration with different external models. Performance comparisons included the CoxKL method (Wang et al., 2023), CoxKL-RIDGE (CoxKL with an additional L2 penalty), NNCox (a neural network Cox model without transfer learning) and were performed on independent testing data. We included an L2 penalty in CoxKL-RIDGE to match the neural network algorithms, that also have an L2 penalty. Each simulation was replicated 100 times.

We simulated survival data from a Cox model with both linear (Setting 1) and non-linear/interaction effects (Setting 2). To evaluate robustness, we introduced varying degrees of domain shift between the external and internal cohorts across three scenarios (E1-E6). The full data-generating processes are detailed in the Appendix. Additional experiments evaluating robustness to uninformative external models, severe domain shifts, and comparisons with stacked methods (Debray et al., 2014a) are detailed in the Appendix.

### 3.1 Linear Simulation:

Figure 1 summarizes the results Simulation Setting I. For different levels of external model quality, the NNCoxKL method consistently outperformed the internal model, demonstrating lower loss (negative log-partial likelihood) and a higher Harrell's C-Index. Higher-quality external information led to more improvements in model performance, as demonstrated by a lower loss and a higher C-Index. Incorporating homogeneous external information led to a lower loss and a higher C-Index, particularly when the quality of the internal cohort was limited. However, in heterogeneous settings with misspecified external scores, the C-Index tended to decrease, though the loss often still improved. As expected, the benefits of external information diminished with larger internal sample

sizes or lower censoring rates. As Figure 1 illustrates, the NNCoxKL model outperformed the internal model in all scenarios when an appropriate penalty $\eta$ was selected. In addition, the optimal range of $\eta$ narrowed with decreasing external information quality.

Figure 1: Comparison of Loss and C-index with different $\eta$ values under simulation setting I. Internal = Deep Learning model based on the internal data only; NNCoxKL = transfer learning model by the proposed method. Panel A and B corresponds to Setting (E1) with a good-quality external model. Panel C and D corresponds to Setting (E2) with a fair-quality external model. Panel E and F corresponds to Setting (E3) with a poor-quality external model. Details of external settings can be found in Section 3. The simulation study was replicated 100 times. Higher C-index indicated better prediction performance.

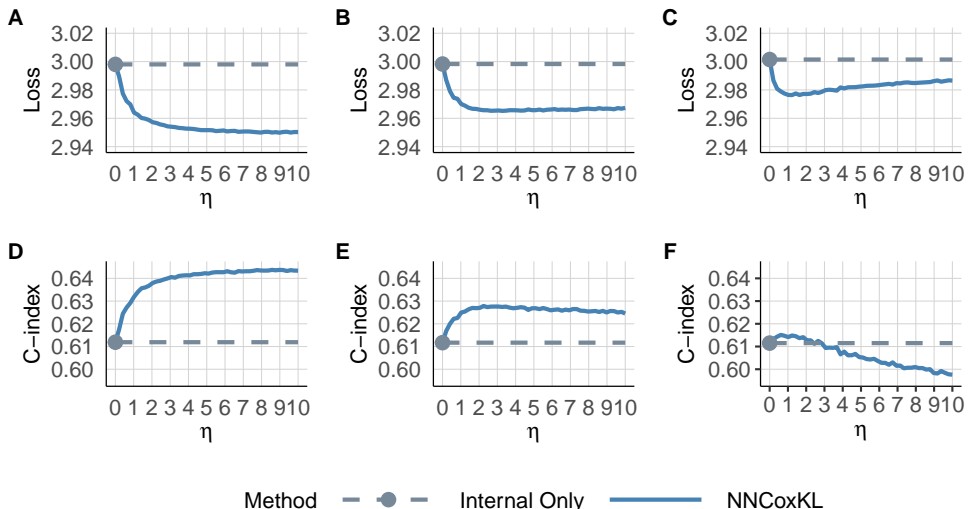

## 3.2 NON-LINEAR SIMULATION:

Figure 2 compares the predictive performance (C-index) of CoxKL, CoxKL-RIDGE and NNCoxKL under different scenarios. The internal dataset had 200 samples, while the external dataset had 3000 samples. A separate testing dataset of 1,000 samples was used for evaluation. "Internal only" refers to models fit exclusively on the internal data only. "High quality" and "Low quality" denote transfer knowledge of external data from simulation settings (E4) and (E5), respectively. In all cases the external model for NNCoxKL is neural network proportional hazards model, and for CoxKL and CoxKL-RIDGE they are linear additive Cox models. Both CoxKL and NNCoxKL exhibited improved performance after transfer learning, regardless of external data quality. Adding a ridge penalty to CoxKL (CoxKL-RIDGE) further enhanced its performance compared to the base model. However, NNCoxKL consistently outperformed CoxKL and CoxKL-RIDGE across all scenarios (internal data only, low-quality, high-quality). This was likely attributable to the non-linear nature of the data generating mechanism, which violated the linearity assumption of CoxKL methods.

Figure 3 illustrates the test loss across epochs for NNCox (with internal data only) and NNCoxKL (with external data) under different sample sizes (500 and 5,000) and settings. Note that, in the context of deep learning, an epoch refers to one complete pass of the training algorithm over the entire training dataset The internal dataset consistently comprised 500 samples, and a separate testing dataset of 1,000 samples was used for evaluation. Notably, NNCox exhibited an initial decrease in loss followed by an increase in later epochs, indicating overfitting. In contrast, NNCoxKL, when integrating information from homogeneous (E4) and heterogeneous (E6) external datasets, reduced overfitting, maintaining a relatively stable loss as the number of epochs increases. For the heterogeneous setting, while a slight increase in loss was observed with larger external sample sizes as epochs increase, this performance remained significantly better than when using internal data only.

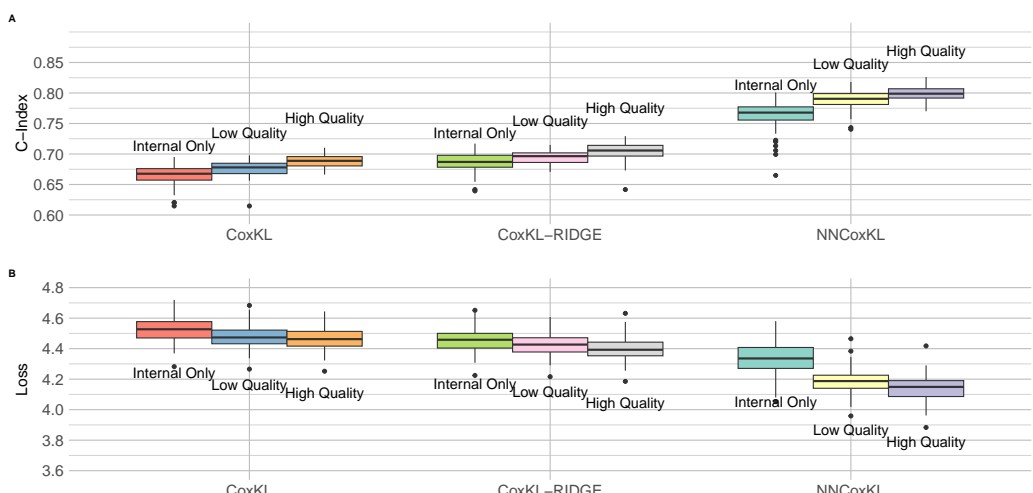

Figure 2: Comparison of prediction performance (C-index and loss) using NNCoxKL, CoxKL and CoxKL-RIDGE. Internal only refers to fitting on internal data only. Low quality refers to transfer knowledge from low quality external data from simulation setting (E5). High quality refers to transfer knowledge frp, high quality external data from simulation setting (E4). CoxKL is the method proposed by Wang et al. (2023), while CoxKL-RIDGE adds a RIDGE penalty to CoxKL. NNCoxKL refers to the proposed transfer learning framework.

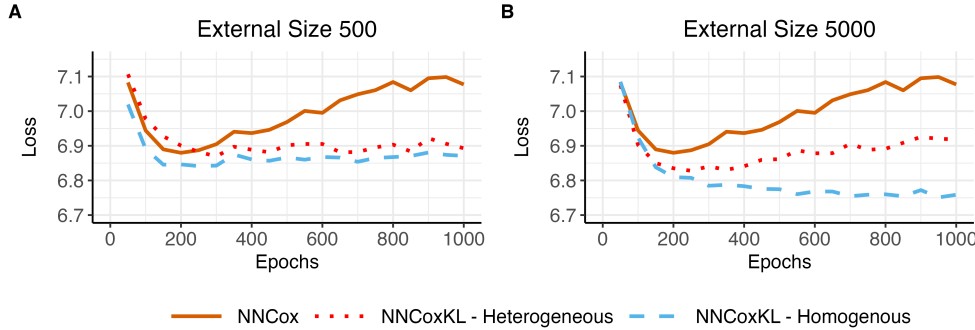

Figure 3: Comparison of loss on an independent testing dataset using NNCox (without transfer learning) and NNCoxKL (with transfer learning) with varying external sample sizes. NNCox shows overfitting issues with increasing loss in later epochs, while NNCoxKL demonstrates more stable loss for both homogeneous setting (E4) and heterogeneous setting (E6), highlighting the benefit of transfer learning in mitigating overfitting, especially with limited internal data.

Consequently, after transfer learning, the performance of the model became less sensitive to the selection of stopping criteria, a challenge often encountered with datasets of moderate size.

## 4 DATA ANALYSIS

Prostate cancer is the second leading cause of cancer death among men in the United States (Siegel et al., 2022). Prognostic predictions are used to guide treatment decisions and improve patient outcomes. The STAR-CAP system (Staging Collaboration for Prostate Cancer), a validated clinical prognostic tool, developed from a study that included 19,684 patients and offered a standardized framework for risk assessment in localized prostate cancer (Dess et al., 2020). However, its points-based nature and potential non-linear relationship with survival outcomes may limit its predictive power in certain contexts. Details of the STAR-CAP system can be found in the Appendix.

We leveraged the NNCoxKL framework to integrate STAR-CAP scores with internal patient-level data from the Michigan Urological Surgery Improvement Collaborative (MUSIC). This data set includes 378 patients diagnosed with prostate cancer and treated with surgery, together with their corresponding STAR-CAP risk group. Our model for the internal data used age, percentage of positive core biopsy (ppc), and pretreatment prostate-specific antigen (PSA) as continuous predictors. All other covariates were treated as categorical, including an additional covariate, the Decipher score (a genomic test used in treatment decision making) with 3 levels, clinical T category with 3 levels, clinical N category with 3 levels, and Gleason grade with 5 levels. The outcome of interest was the time to biochemical recurrence (BCR), defined as at least two PSA values of 0.2 ng / ml or greater after radical prostatectomy.

By tranferring knowledge from the STAR-CAP risk groups with the individual-level data from MU-SIC, we aimed to leverage the strengths of both approaches. The STAR-CAP system provides a robust and standardized risk assessment framework, while the NNCoxKL model allows for the capture of complex, nonlinear relationships between clinical factors, genomics information (decipher score), and BCR outcomes. As illustrated in Figure 4, incorporating STAR-CAP scores into the NNCoxKL model enhanced predictive performance by achieving a higher C-index and a lower Loss compared to using MUSIC data alone.

Figure 4: Results of predictive performance for biochemical recurrence (BCR). The censoring rate is 81.37%. The internal data was randomly split into a training set (80%) and a testing set (20%). Results are evaluated on the independent testing set. This random partitioning and evaluation was repeated 20 times Internal only refers to fitting on internal data only without STAR-CAP information. NNCoxKL refers to the proposed transfer learning framework. Optimal $\eta$ was selected to be around 1 via cross-validation. Panels (A) and (C) show the improvement in C-index and Loss, across different random training/testing splits. Panel (B) shows the C-index across different values of the transfer learning tuning parameter $\eta$, while Panel (D) presents the loss across different $\eta$ values.

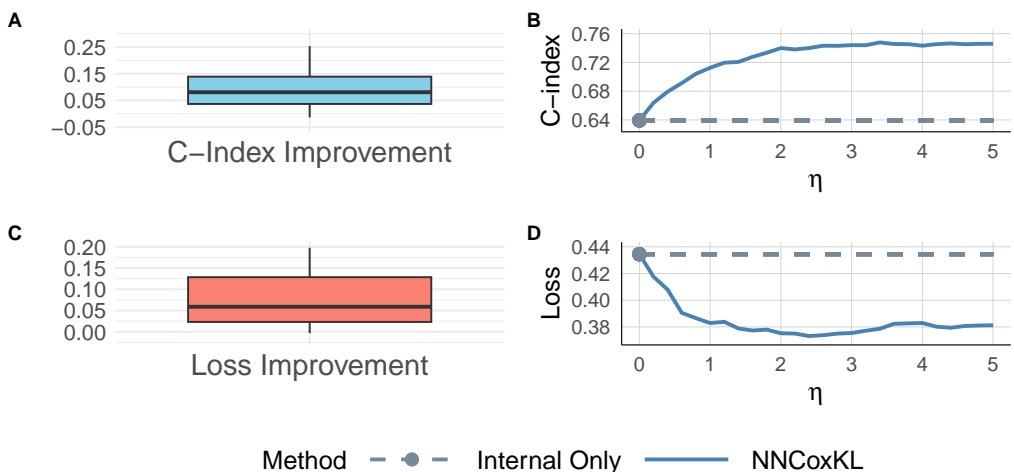

## 5 DISCUSSION

In this paper, we have introduced NNCoxKL, a flexible deep learning framework designed to integrate external risk models with internal time-to-event data for enhanced prognostic prediction. Our approach is adaptable to both homogeneous and heterogeneous settings, accommodating various types of external risk score, including those derived from nontraditional approaches such as the STAR-CAP system. By leveraging the power of deep neural networks, NNCoxKL effectively captures complex non-linear relationships between covariates and survival outcomes, leading to improved predictive accuracy.

Through simulation studies and the three benchmark datasets, we have demonstrated the superior performance of NNCoxKL compared to traditional methods, particularly in scenarios with limited sample sizes or complex covariate relationships. The real-world application to the prognosis of prostate cancer further highlights the practical utility of our framework. By integrating STAR-CAP risk groups with individual-level data from the MUSIC cohort, we were able to improve the prediction of biochemical recurrence, showcasing the potential of NNCoxKL.

While in this work our primary focus is proportional hazards modeling, the NNCoxKL framework can be naturally extended to model non-proportional hazards. By allowing the network to output a time-dependent risk function $r(\mathbf{Z}, t)$ as used in Cox-Time (Kvamme et al., 2019), the KL-divergence loss can be computed dynamically at each event time. Although this increases the computational cost during training, it remains feasible within standard mini-batch optimization pipelines.

In addition, although our empirical studies focus on a single event type, NNCoxKL naturally extends to competing risks. In the appendix (Section A.2), we outline a cause-specific hazard formulation in which separate neural networks are trained for each event type and the generalized KL divergence is applied to align internal predictions with external risk information for the event of interest.

Furthermore, the proposed method can be extended to incorporate multiple external sources in two ways. First, NNCoxKL can be directly extended by adding multiple KL-based penalty terms, each accommodating the discrepancy between the internal data and one external data source. This allows simultaneous incorporation of multiple external datasets while accounting for the heterogeneity of each source individually. Alternatively, multiple external models can be combined using existing model aggregation strategies (Debray et al., 2014a; LECUÉ & RIGOLLET, 2014), and the resulting aggregated model can then be incorporated as the external model within the NNCoxKL framework.

Although our proposed framework offers significant advantages, it also presents some limitations. As with any transfer-learning approach, the benefit of NNCoxKL depends on the relevance and fidelity of the external information. Poor or misaligned sources naturally provide more limited improvement. On the other hand, because the transfer strength is adaptively tuned through $\eta$, the method down-weights unhelpful external information, allowing stable performance and still offering potential gains even when the external model is not well aligned. Furthermore, although NNCoxKL accommodates heterogeneity across datasets, computing the external risk score $\tilde{r}(\boldsymbol{Z}_i)$ requires the internal dataset to contain the relevant covariates or their derivable counterparts. When these requisite features are absent, the method cannot be applied.

Moreover, by introducing deep neural networks, NNCoxKL trades some of the interpretability of classical Cox models for greater flexibility. In practice, this loss can be partially mitigated by post hoc tools such as variable-importance analyses and partial dependence plots, or by distilling the learned NNCoxKL predictor into a simpler surrogate model (e.g., a Cox or tree-based model) using knowledge-distillation techniques (Hinton et al., 2015; Guo et al., 2017).

Future research could explore several extensions of the NNCoxKL framework. One potential direction is to investigate alternative methods instead of cross-validation for selecting the integration parameter, such as by using information criteria. Another potential area is to expand the framework to accommodate multiple external risk models simultaneously, which could lead to further improvements in predictive accuracy. Furthermore, exploring the application of NNCoxKL to other types of survival outcomes and clinical settings would be valuable.

In conclusion, the proposed NNCoxKL offers a flexible and powerful framework for integrating external risk models with internal time-to-event data, enhancing prognostic prediction in survival analysis, particularly in challenging scenarios with limited data or complex relationships.

## REPRODUCIBILITY STATEMENT

We provide the information needed to reproduce all results. The full learning objective, model class, and training procedure are specified in Section 2 (including the penalized partial likelihood with the generalized KL term and Proposition 1), with notation and implementation details given in Eqs.1 - 5 and the surrounding text. We describe the experimental design and evaluation protocol for simulations in Section 3, and for real-data analyses in Section 4. Figures report performance on held-out test sets and, where applicable, averages across repeated random splits; the MUSIC+STAR-CAP analysis also reports the cross-validated selection of $\eta$. The selection of the $\eta$ employed 5-fold cross-validation with the $V\&VH$ cross-validated partial likelihood (Verweij & Van Houwelingen, 1993) as the performance metric. To facilitate replication, we will release an anonymized repository containing: (i) scripts to generate all simulated datasets and reproduce tables/figures; (ii) preprocessing and split scripts for each real dataset; and (iii) training/evaluation code for NNCoxKL and baselines with fixed random seeds and documented hyperparameters. The appendix includes a complete proof of Proposition 1.

## AI USAGE DISCLOSURE

The authors used an AI-powered language tool for copy editing purposes, including grammar checks and refinement of wording for clarity.

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

# A  APPENDIX

## A.1  PROOF OF PROPOSITION 1

The generalized KL is given by

$$
\mathrm{d}(\mathbf{p}_k \| \mathbf{q}_k) = \sum_{i \in R_k} \mathbf{p}_k(i) \log \frac{\mathbf{p}_k(i)}{\mathbf{q}_k(i)}
$$

$$
= -\sum_{i=1}^{n} \frac{Y_i(t_k) \exp\{\tilde{r}(\mathbf{Z}_i)\}}{\sum_{j=1}^{n} Y_j(t_k) \exp\{\tilde{r}(\mathbf{Z}_j)\}} \left[ r(\mathbf{Z}_i, \boldsymbol{\beta}) - \log \left\{ \sum_{j=1}^{n} Y_j(t_k) \exp\{r(\mathbf{Z}_j, \boldsymbol{\beta})\} \right\} \right] + \Psi_k,
$$

where $\Psi_k$ is a constant not involving $\boldsymbol{\beta}$. The accumulated generalized KL is

$$
\mathrm{D}(\tilde{\mathbf{r}} \| \mathbf{r}) = \sum_{k=1}^{K} \mathrm{d}(\mathbf{p}_k \| \mathbf{q}_k)
$$

$$
= -\sum_{k=1}^{K} \sum_{i=1}^{n} \frac{Y_i(t_k) \exp\{\tilde{r}(\mathbf{Z}_i)\}}{\sum_{j=1}^{n} Y_j(t_k) \exp\{\tilde{r}(\mathbf{Z}_j)\}} \left[ r(\mathbf{Z}_i, \boldsymbol{\beta}) - \log \left\{ \sum_{j=1}^{n} Y_j(t_k) \exp\{r(\mathbf{Z}_j, \boldsymbol{\beta})\} \right\} \right] + \Psi,
$$

where $\Psi = \sum_{k=1}^{K} \Psi_k$ is a constant not involving $\boldsymbol{\beta}$. Ignoring such constant terms, the penalized objective function is then given by:

$$
\ell_\eta = \sum_{k=1}^{K} \sum_{i=1}^{n} \left[ \delta_i(t_k) + \frac{\eta Y_i(t_k) \exp\{\tilde{r}(\mathbf{Z}_i)\}}{\sum_{j=1}^{n} Y_j(t_k) \exp\{\tilde{r}(\mathbf{Z}_j)\}} \right] \left[ r(\mathbf{Z}_i, \boldsymbol{\beta}) - \log\left\{ \sum_{j=1}^{n} Y_j(t_k) \exp\{r(\mathbf{Z}_j, \boldsymbol{\beta})\} \right\} \right]
$$

$$
= \sum_{i=1}^{n} \left[ \sum_{k=1}^{K} \delta_i(t_k) \cdot r(\mathbf{Z}_i, \boldsymbol{\beta}) + \sum_{k=1}^{K} \frac{\eta Y_i(t_k) \exp\{\tilde{r}(\mathbf{Z}_i)\}}{\sum_{j=1}^{n} Y_j(t_k) \exp\{\tilde{r}(\mathbf{Z}_j)\}} \cdot r(\mathbf{Z}_i, \boldsymbol{\beta}) \right]
$$

$$
- \sum_{k=1}^{K} \left[ \sum_{i=1}^{n} \delta_i(t_k) + \sum_{i=1}^{n} \frac{\eta Y_i(t_k) \exp\{\tilde{r}(\mathbf{Z}_i)\}}{\sum_{j=1}^{n} Y_j(t_k) \exp\{\tilde{r}(\mathbf{Z}_j)\}} \right] \log\left\{ \sum_{j=1}^{n} Y_j(t_k) \exp\{r(\mathbf{Z}_j, \boldsymbol{\beta})\} \right\}
$$

$$
= \sum_{i=1}^{n} \sum_{k=1}^{K} \left[ \delta_i(t_k) + \frac{\eta Y_i(t_k) \exp\{\tilde{r}(\mathbf{Z}_i)\}}{\sum_{j=1}^{n} Y_j(t_k) \exp\{\tilde{r}(\mathbf{Z}_j)\}} \right] r(\mathbf{Z}_i, \boldsymbol{\beta})
$$

$$
- \sum_{k=1}^{K} (1 + \eta) \log\left\{ \sum_{j=1}^{n} Y_j(t_k) \exp\{r(\mathbf{Z}_j, \boldsymbol{\beta})\} \right\}
$$

$$
= \sum_{i=1}^{n} \left[ (\delta_i + \eta \tilde{\delta}_i) r(\mathbf{Z}_i, \boldsymbol{\beta}) - (1 + \eta) \delta_i \log\left\{ \sum_{j=1}^{n} Y_j(t_i) \exp\{r(\mathbf{Z}_j, \boldsymbol{\beta})\} \right\} \right]
$$

$$
\propto \sum_{i=1}^{n} \left[ \frac{\delta_i + \eta \tilde{\delta}_i}{1 + \eta} r(\mathbf{Z}_i, \boldsymbol{\beta}) - \delta_i \log\left\{ \sum_{j=1}^{n} Y_j(t_i) \exp\{r(\mathbf{Z}_j, \boldsymbol{\beta})\} \right\} \right],
$$

where $\delta_i(t_k) = I(T_i \le C_i, T_i = t_k)$ is the event indicator for subject $i$ at time $t_k$, and

$$
\tilde{\delta}_i = \sum_{k=1}^{K} \frac{Y_i(t_k) \exp\{\tilde{r}(\mathbf{Z}_i)\}}{\sum_{j=1}^{n} Y_j(t_k) \exp\{\tilde{r}(\mathbf{Z}_j)\}}.
$$

Note that for the 2nd equation to the 3rd, we have $\sum_{i=1}^{n} \delta_i(t_k) = 1$, assuming no ties. To transition from the 3rd equation to the 4th, we apply the following relationship:

$$
\sum_{k=1}^{K} (1 + \eta) \log\left\{ \sum_{j=1}^{n} Y_j(t_k) \exp\{r(\mathbf{Z}_j, \boldsymbol{\beta})\} \right\} = \sum_{i=1}^{n} (1 + \eta) \delta_i \log\left\{ \sum_{j=1}^{n} Y_j(t_i) \exp\{r(\mathbf{Z}_j, \boldsymbol{\beta})\} \right\}.
$$

### A.2 COMPETING RISK EXTENSION

To incorporate competing risks, our proposed framework can be naturally extended through cause-specific hazard modeling (Kalbfleisch & Prentice, 2011), which is a natural extension of the Cox model. Specifically, the cause-specific hazard for cause $m$ assumes a multiplicative effect:

$$
\lambda_m(t \mid \mathbf{Z}_i) = \lambda_{0m}(t) \exp\{r_m(\mathbf{Z}_i, \boldsymbol{\beta}_m)\},
$$

where both the baseline hazards $\lambda_{0m}(t)$ and the neural network can differ across causes. The resulting cause-specific hazard ratio quantifies the effect of a covariate on the hazard of a specific event type. Because the NNCoxKL framework is constructed from the partial likelihood, it supports direct training of separate cause-specific models without requiring architectural changes. To describe the probabilities of transitioning to competing events, the corresponding cumulative incidence functions (CIFs) can be computed in a straightforward manner from the estimated cause-specific hazards (Kalbfleisch & Prentice, 2011).

### A.3 TIE HANDLING UNDER THE BRESLOW APPROXIMATION

In many applications, event times are tied because they are recorded on visit schedules or rounded to calendar units. We handle such ties using the standard Breslow approximation to the Cox partial likelihood, extended to the general log–risk function $r(\mathbf{z}_i, \boldsymbol{\beta})$.

Following the notation in Section 2.1, let $t_1 < \cdots < t_K$ denote the unique failure times in the internal cohort.

For each unique failure time $t_k$, let $D_k$ denote the set of individuals who failed at that time, and let $d_k = |D_k|$ be the number of tied events. Specifically:

$$D_k = \{i : X_i = t_k, \delta_i = 1\}.$$

The risk set just prior to time $t_k$, denoted as $R_k$, consists of all individuals still under observation:

$$R_k = \{j : X_j \geq t_k\} = \{j : Y_j(t_k) = 1\}.$$

**Internal model.** Under the Breslow approximation, the denominator of the partial likelihood is the same for all individuals failing at time $t_k$. The log-partial likelihood for the internal model, $\ell(\boldsymbol{\beta})$, is adapted from Eq. 1 as follows:

$$\ell^{\mathrm{B}}(\boldsymbol{\beta}) = \sum_{k=1}^{K} \left\{ \sum_{i \in D_k} r(\mathbf{Z}_i, \boldsymbol{\beta}) - d_k \log \left( \sum_{j \in R_k} \exp\{r(\mathbf{Z}_j, \boldsymbol{\beta})\} \right) \right\}. \tag{6}$$

When there are no ties (i.e., $d_k = 1$ for all $k$), this reduces exactly to the standard log-partial likelihood in Eq. 1.

**External model and KL divergence.** We utilize the external risk scores $\tilde{r}(\mathbf{Z}_i)$ as defined in Section 2.2. To handle ties in the transfer learning component, we consider the distribution of risk over the risk set $R_k$. We define the teacher ($\mathbf{p}_k$) and student ($\mathbf{q}_k$) distributions for an individual $j \in R_k$ as:

$$\mathbf{p}_k(j) = \frac{\exp\{\tilde{r}(\mathbf{Z}_j)\}}{\sum_{\ell \in R_k} \exp\{\tilde{r}(\mathbf{Z}_\ell)\}}, \qquad \mathbf{q}_k(j) = \frac{\exp\{r(\mathbf{Z}_j, \boldsymbol{\beta})\}}{\sum_{\ell \in R_k} \exp\{r(\mathbf{Z}_\ell, \boldsymbol{\beta})\}}.$$

The generalized KL divergence at time $t_k$, weighted by the number of ties, is given by:

$$\mathrm{d}_k(\mathbf{p}_k \| \mathbf{q}_k) = d_k \sum_{j \in R_k} \mathbf{p}_k(j) \log \frac{\mathbf{p}_k(j)}{\mathbf{q}_k(j)}. \tag{7}$$

This formulation ensures that times with a higher density of events contribute proportionally to the divergence penalty. To compute this efficiently, let $A_k = \sum_{j \in R_k} \exp\{\tilde{r}(\mathbf{Z}_j)\}$ and $B_k(\boldsymbol{\beta}) = \sum_{j \in R_k} \exp\{r(\mathbf{Z}_j, \boldsymbol{\beta})\}$. The divergence at $t_k$ simplifies to:

$$\mathrm{d}_k(\mathbf{p}_k \| \mathbf{q}_k) = d_k \left[ \frac{1}{A_k} \sum_{j \in R_k} \exp\{\tilde{r}(\mathbf{Z}_j)\} \big(\tilde{r}(\mathbf{Z}_j) - r(\mathbf{Z}_j, \boldsymbol{\beta})\big) - \log A_k + \log B_k(\boldsymbol{\beta}) \right]. \tag{8}$$

The accumulated generalized KL divergence across all unique failure times is:

$$\mathrm{D}^{\mathrm{B}}(\tilde{\mathbf{r}} \| \mathbf{r}) = \sum_{k=1}^{K} \mathrm{d}_k(\mathbf{p}_k \| \mathbf{q}_k). \tag{9}$$

Finally, the penalized log-partial likelihood with Breslow tie handling becomes:

$$\ell_\eta^{\mathrm{B}}(\boldsymbol{\beta}) = \ell^{\mathrm{B}}(\boldsymbol{\beta}) - \eta \, \mathrm{D}^{\mathrm{B}}(\tilde{\mathbf{r}} \| \mathbf{r}). \tag{10}$$

This formulation is computationally efficient as it relies on aggregated sums over the risk sets $R_k$, addressing concerns regarding the computational cost of ranking-based losses in deep learning frameworks.

### A.3.1 EMPIRICAL VALIDATION OF TIE HANDLING

To demonstrate the robustness of the proposed tie-handling implementation, we performed an ablation study using the METABRIC dataset. We utilized the standard split: internal (10%, $n = 190$), external (70%, $n = 1333$), and testing (20%, $n = 381$).

We compared the performance of NNCoxKL under two conditions:

1. Original: Using the original event times (1686 unique times).

2. Rounded (Tied): We rounded all event times up to the nearest integer. This artificially induced a high degree of ties, reducing the number of unique event times from 1686 to 303.

As shown in Table A.1, the NNCoxKL framework utilizing the Breslow approximation maintains consistent predictive performance (C-index) and Loss, even when the number of unique event times is drastically reduced. This confirms that the generalized KL divergence derived in Eq. 8 effectively handles risk set calculations in the presence of tied data.

Table A.1: Impact of tie-handling on NNCoxKL performance (METABRIC dataset). The 'Rounded' setting creates heavy ties, reducing unique event times by 82%. The performance remains stable, validating the Breslow approximation strategy.

| Condition | Unique Event Times | Loss | C-index |
|---|---|---|---|
| Original Data | 1686 | 2.943 | 0.634 |
| Rounded Data (Ties) | 303 | 2.949 | 0.633 |

### A.4 EXTENSION TO NON-PROPORTIONAL HAZARDS MODELING

The NNCoxKL framework, while primarily presented in the context of the proportional hazards (PH) model, can be naturally and flexibly extended to accommodate time-varying effects and thus address the non-proportional hazards assumption. This extension is achieved by allowing the neural network to output a time-dependent log-relative risk function, $r(\mathbf{Z}_i, t, \boldsymbol{\beta})$, similar to the Cox-Time approach (Kvamme et al., 2019).

Following Cox-Time (Kvamme et al., 2019), we allow the network to output a time-dependent risk function

$$\lambda(t \mid \mathbf{Z}_i) = \lambda_0(t) \exp\{r(\mathbf{Z}_i, t; \boldsymbol{\beta})\},$$

where $r(\mathbf{Z}_i, t; \boldsymbol{\beta})$ is implemented by a neural network that takes both covariates and time as input. Using the notation from Section 2.1, let $t_1 < \cdots < t_K$ denote the unique event times, $Y_j(t_k)$ the at-risk indicator, and $\delta_i(t_k)$ the event indicator for subject $i$ at time $t_k$. The Cox partial likelihood in this time-varying setting becomes

$$\ell^{\mathrm{TV}}(\boldsymbol{\beta}) = \sum_{k=1}^{K} \sum_{i=1}^{n} \delta_i(t_k) \Big[ r(\mathbf{Z}_i, t_k; \boldsymbol{\beta}) - \log\Big\{ \sum_{j=1}^{n} Y_j(t_k) \exp\{r(\mathbf{Z}_j, t_k; \boldsymbol{\beta})\} \Big\} \Big]. \quad (11)$$

The KL term is modified analogously by defining teacher and student Plackett–Luce probabilities at each event time:

$$\mathbf{p}_k(i) = \frac{\exp\{\tilde{r}(\mathbf{Z}_i, t_k)\}}{\sum_{j \in R_k} \exp\{\tilde{r}(\mathbf{Z}_j, t_k)\}}, \qquad \mathbf{q}_k(i) = \frac{\exp\{r(\mathbf{Z}_i, t_k; \boldsymbol{\beta})\}}{\sum_{j \in R_k} \exp\{r(\mathbf{Z}_j, t_k; \boldsymbol{\beta})\}},$$

where $R_k = \{j : Y_j(t_k) = 1\}$ is the risk set just prior to $t_k$, and $\tilde{r}(\mathbf{Z}_i, t_k)$ denotes a time-dependent external risk score (when available). The accumulated divergence is then

$$D^{\mathrm{TV}}(\tilde{\mathbf{r}} \,\|\, \mathbf{r}) = \sum_{k=1}^{K} d\big(\mathbf{p}_k \,\|\, \mathbf{q}_k\big),$$

with $d(\cdot \| \cdot)$ as in Eq. equation 2. The time-varying NNCoxKL objective is

$$\ell_\eta^{\mathrm{TV}}(\boldsymbol{\beta}) = \ell^{\mathrm{TV}}(\boldsymbol{\beta}) - \eta \, D^{\mathrm{TV}}(\tilde{\mathbf{r}} \,\|\, \mathbf{r}). \quad (12)$$

### A.5 SIMULATION DETAILS

#### A.5.1 LINEAR SIMULATION SETTINGS

For simulation setting 1, six covariates $(\mathbf{Z}_1, \ldots, \mathbf{Z}_6)$ were used. $\mathbf{Z}_1$ and $\mathbf{Z}_2$ were continuous, generated from a multivariate normal distribution (mean zero, unit variance) with AR1 correlation (parameter 0.5). $\mathbf{Z}_3$ and $\mathbf{Z}_4$ were binary with $\Pr(\mathbf{Z}_i = 1) = 0.5$. To simulate distributional changes between

internal and external cohorts, $\mathbf{Z}_5$ and $\mathbf{Z}_6$ were continuous, normally distributed (unit variance) with means of $2\mathbf{Z}^\ell$ and $-2\mathbf{Z}^\ell$, respectively, where $\mathbf{Z}^\ell$ was a latent binary variable ($\Pr(\mathbf{Z}_i^\ell = 1) = p_\ell$). We then generated survival times from a Cox model:

$$\lambda(t|\mathbf{Z}_{i1}, \ldots, \mathbf{Z}_{i6}) = 2t \times \exp(\beta_1\mathbf{Z}_{i1} + \beta_2\mathbf{Z}_{i2} + \beta_3\mathbf{Z}_{i3} + \beta_4\mathbf{Z}_{i4} + \beta_5\mathbf{Z}_{i5} + \beta_6\mathbf{Z}_{i6}) \qquad (13)$$

with $\boldsymbol{\beta}^I = (0.3, -0.3, 0.3, -0.3, 0.3, -0.3)^\top$. The censoring times were uniformly distributed with varying bounds for different censoring rates. The internal dataset had a sample size of 125, while the external dataset had a sample size of 2,000. Model performance was evaluated on a separate, independent testing dataset with a sample size of 1,000. For the internal cohort, $p_\ell^I = 1$. We explored three external model settings:

(E1). The covariate distribution of the external population was the same as the internal cohort; that is, $p_l^E = 1$; and the external model was the true model: $Z^E = (Z_1, Z_2, Z_3, Z_4, Z_5, Z_6)^T$.

(E2). The covariate distribution of the external population was slightly different from the internal cohort with $p_l^E = 0.5$; and the external model was a misspecified model: $Z^E = (Z_1, Z_3, Z_5, Z_6)^T$.

(E3). The covariate distribution of the external population was completely different from the internal cohort with $p_l^E = 0$; and the external model was a misspecified model: $Z^E = (Z_1, Z_5)^T$.

### A.5.2 Non-linear Simulation Settings

In simulation setting 2, we included 16 covariates $\mathbf{Z}_1, \ldots, \mathbf{Z}_{16}$. Except for $\mathbf{Z}_3$ and $\mathbf{Z}_4$, all covariates were continuous, generated from a multivariate normal distribution (mean zero, unit variance) with AR1 correlation (parameter 0.5), while $\mathbf{Z}_3$ and $\mathbf{Z}_4$ were binary with $\Pr(\mathbf{Z}_i = 1) = 0.5$. In addition to the main effect, we considered non-linear effects $\sin(2\pi\mathbf{Z}_1)$ and $\exp(-\mathbf{Z}_2)$, quadratic terms $\mathbf{Z}_1^2$, $\mathbf{Z}_2^2$, $\mathbf{Z}_3^2$ and $\mathbf{Z}_4^2$, and interaction terms $\mathbf{Z}_1\mathbf{Z}_2$, $\mathbf{Z}_2\mathbf{Z}_3$, $\mathbf{Z}_1\mathbf{Z}_3$ and $\mathbf{Z}_2\mathbf{Z}_4$. Since in practice we may collect some variables not included in the true model, we simulated twelve additional variables $\mathbf{Z}_5, \ldots, \mathbf{Z}_{16}$, which did not contribute to the true underlying generating distribution of the event time. The true generating model for the internal data was a proportional hazards model with a hazard function of

$$\lambda(t|\mathbf{Z}_{i1}, \ldots, \mathbf{Z}_{i4}) = t \times \exp(\sin(2\pi\mathbf{Z}_1) + \exp(-\mathbf{Z}_2) + \beta_1(\mathbf{Z}_1^2 + \mathbf{Z}_2\mathbf{Z}_3)$$
$$+ \beta_2(\mathbf{Z}_2^2) + \beta_3(\mathbf{Z}_3^2 + \mathbf{Z}_1\mathbf{Z}_2) + \beta_4(\mathbf{Z}_4^2) + \beta_5\mathbf{Z}_1\mathbf{Z}_3 + \beta_6\mathbf{Z}_2\mathbf{Z}_4)$$

with $\boldsymbol{\beta} = (0.3, -0.3, 0.3, -0.3, 0.3, -0.3)^\top$.

In a comprehensive evaluation of the proposed NNCoxKL, we explored the following external model configurations, all constructed using neural network architectures:

- (E4) Homogeneous (High Quality): The external model was constructed using the same neural network architecture and hyperparameter set as for the internal model.

- (E5) Homogeneous (Low Quality): The external covariate distribution mirrored the internal cohort with different censoring distribution, resulting in an event rate of 42.2%.

- (E6) Heterogeneous: The external covariate set was reduced, represented as $Z^E = (Z_1, Z_2, Z_3)^\top$.

### A.6 Additional Simulation Setting: External Information Using Risk Group Classification

In this subsection, we generated external information to mimic the situation with the STAR-CAP system. We included seven variables, $Z_1, \ldots, Z_7$ derived from 5 covariates. We first generated a bivariate normal distribution for covariates $Z_1$ and $Z_6$ with mean vector 0, unit variance and AR1 correlation (parameter 0.5). We then dichotomized the $Z_1$ and $Z_6$ into binary variables (0 or 1), with the probability of being zero uniformly varying from 0.25 to 0.35. We then created two categorical variables with three distinct levels and encoded them using dummy variables $Z_2$ and $Z_3$, $Z_4$ and $Z_5$ separately. $Z_7$ was continuous and generated from a normal distribution with mean 0, 1 and 2, corresponding to the values of $(Z_2 = 0, Z_3 = 0)$, $(Z_2 = 1, Z_3 = 0)$ and $(Z_2 = 0, Z_3 = 1)$. The true

generating model for the internal data was a proportional hazards model with the following hazard function:

$$\lambda(t|\mathbf{Z}_{i1}, \ldots. \mathbf{Z}_{i7}) = 2t \times \exp(\beta_1 \mathbf{Z}_{i1} + \beta_2 \mathbf{Z}_{i2} + \beta_3 \mathbf{Z}_{i3} + \beta_4 \mathbf{Z}_{i4} + \beta_5 \mathbf{Z}_{i5}$$
$$+ \beta_6 \mathbf{Z}_{i6} + \beta_7 (\mathbf{Z}_{i1}\mathbf{Z}_{i6}) + \beta_8 \exp(-\mathbf{Z}_{i7}^2)),$$

with $\boldsymbol{\beta} = (0.25, 0.25, 0.5, 0.5, 0.75, 0.25, 0.5, -0.5)$. To define the external risk groups, we assigned points based on specific covariates: 1 point was assigned if $Z_1 = 1$; 1 point if $Z_2 = 1$; 2 points if $Z_3 = 1$; 2 points if $Z_4 = 1$; and 3 points if $Z_5 = 1$, and we added these points for each individual.

Figure A.1 presents the prediction performance of the proposed NNCoxKL framework versus the stacked method of Debray et al. (2014b). Panel A shows the C-index, and Panel B shows the prediction loss, both evaluated across various values of the tuning parameter $\eta$. By increasing the weight given to external information (i.e. higher $\eta$), the NNCoxKL framework achieved higher C-index and lower prediction loss than the stacked method. This indicates that NNCoxKL effectively leverages the points-based external data to improve prediction accuracy. Notably, the NNCoxKL framework outperforms the use of internal data alone (solid line), while the stacked method shows limited improvement. These results demonstrate the advantage of the NNCoxKL framework in transferring knowledge from points-based external information for enhanced prediction performance.

Figure A.1: Comparison of prediction performance between the proposed NNCoxKL framework and the stacked method of (Debray et al., 2014b). (A) C-index. (B) Loss. Performance is evaluated across different values of the tuning parameter $\eta$. The solid line represents results using only internal data, while the long-dashed line represents results for the stacked method. The simulation setting is based on Section A.6.

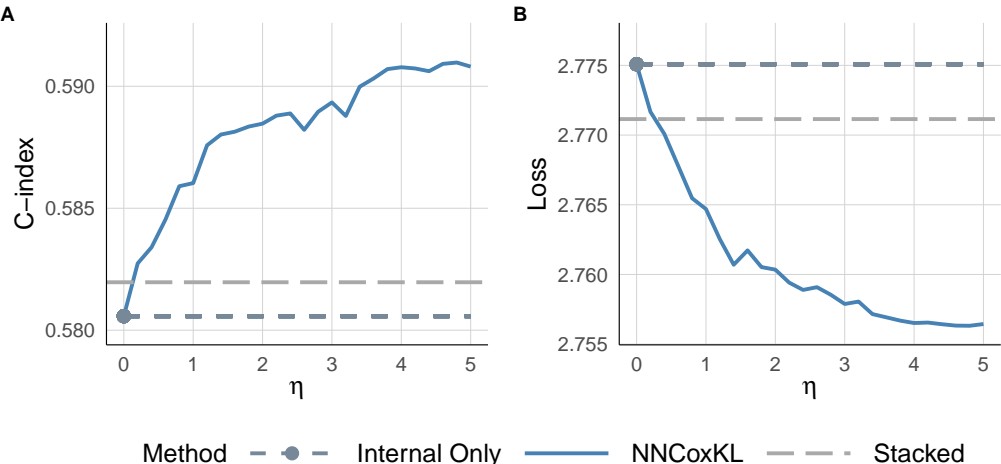

A.7   ADDITIONAL SIMULATION SETTING: INTERNAL DATA AND EXTERNAL MODEL COME FROM DIFFERENT COVARIATE DISTRIBUTIONS.

We conducted additional simulations to evaluate the framework's robustness under domain shift, specifically when the marginal prevalence of features differs significantly between the internal and external populations.

We considered 10 binary predictors, $\mathbf{Z} = \{Z_1, \ldots, Z_{10}\}$, generated from independent Bernoulli distributions with parameter $p$, such that $P(Z_j = 1) = p$ for $j = 1, \ldots, 10$. Failure times were generated from an exponential model with hazard rate $\lambda(t|\mathbf{Z}) = \exp(\eta(\mathbf{Z}))$. The log-hazard $\eta(\mathbf{Z})$ was modeled as a combination of non-linear effects (applied to the binary values), linear effects, and random noise:

$$\eta(\mathbf{Z}) = f_1(\mathbf{Z}) + f_2(\mathbf{Z}) + \epsilon, \tag{14}$$

where $\epsilon \sim \mathcal{N}(0, 1.5^2)$. The non-linear component $f_1$ captures complex interactions among the first five covariates:

$$f_1(\mathbf{Z}) = \sin(0.5\pi Z_1) + \exp(-Z_2) + 0.2 \times \Big[ -1.5(Z_1^2 + Z_2 Z_3) - 1.0(Z_2^2)$$
$$-1.8(Z_3^2 + Z_1 Z_2) - 0.9(Z_4^2)$$
$$+ 0.6(Z_1 Z_3) - 0.2(Z_2 Z_4 + Z_2 Z_5)\Big]. \quad (15)$$

The linear component $f_2$ is defined as:

$$f_2(\mathbf{Z}) = -0.3Z_6 + 0.3Z_7 - 0.2Z_8 + 0.5Z_9 + 0.3Z_{10}. \quad (16)$$

The internal dataset ($N = 200$) was generated using the full log-hazard specification with sample size of 200. The covariates were generated with $p = 0.5$, representing a maximum variance scenario for binary predictors. The censoring rate was approximately 40%.

For the external dataset, we introduced two sources of heterogeneity. First We simplified the risk mechanism to mimic a scenario where the external population follows a different data generating process. The quadratic and interaction terms in $f_1$ were removed, resulting in the following simplified log-hazard:

$$\eta_{ext}(\mathbf{Z}) = \sin(0.5\pi Z_1) + \exp(-Z_2) + f_2(\mathbf{Z}) + \epsilon. \quad (17)$$

Second, we modulated the Bernoulli parameter $p$ to create two experimental settings representing different degrees of distributional divergence: Covariate Shift: We modulated the Bernoulli parameter $p$ to create two experimental settings:

- Setting (r1) [Moderate Shift]: $p = 0.3$, representing a moderate deviation from the internal data ($p = 0.5$).
- Setting (r2) [Severe Shift]: $p = 0.9$, representing a severe deviation where the covariate distribution is highly skewed.

The censoring rate of the internal data was approximately 40%, and the sample size of the internal data was set as 200. The external risk scores were derived by training a deep neural network on this external dataset with sample size 500, utilizing the same network architecture as the internal model. Simulations were replicated 100 times.

Figure A.7 displays the boxplot across both shift scenarios using Harrell's C-index, Prediction Loss, Integrated Brier Score (IBS), and Time-dependent AUC. In the case of moderate shift (Setting r1), the external model achieves performance comparable to the internal-only model. NNCoxKL successfully leverages this external signal, yielding the highest C-index and lowest Loss, significantly outperforming the baselines. Under severe shift (Setting r2, $p = 0.9$), the external model performs poorly, yielding a C-index lower than the internal-only model. Despite this high risk for negative transfer, NNCoxKL still outperforms the internal-only model. This demonstrates that the generalized KL penalty allows the model to selectively integrate ranking information, robustly handling scenarios where the external source is significantly misaligned.

Furthermore, we compared NNCoxKL against the Stacked approach Debray et al. (2014b), where the external risk score is included as an additional input covariate in the internal model. We observed that the Stacked method provided negligible improvement, particularly in Setting (r2). When the covariate distribution differs substantially, the direct mapping of external scores to internal risk becomes unreliable. In contrast, NNCoxKL demonstrated superior resilience to this extreme shift.

### A.8 ADDITIONAL REAL WORLD DATASETS

We also performed experiments on three widely used machine learning datasets: the SUPPORT, the Molecular Taxonomy of Breast Cancer International Consortium (METABRIC), and the Rotterdam tumor bank and German Breast Cancer Study Group, (GBSG, Schumacher et al. (1994)). The sample sizes, percent censoring and number of covariates for each dataset were 8873, 32% and 14 for SUPPORT, were 1904, 42% and 9 for METABRIC, and 2232, 43% and 8 for GBSG. To evaluate the performance of transfer learning, we randomly split each dataset into internal, external, and testing

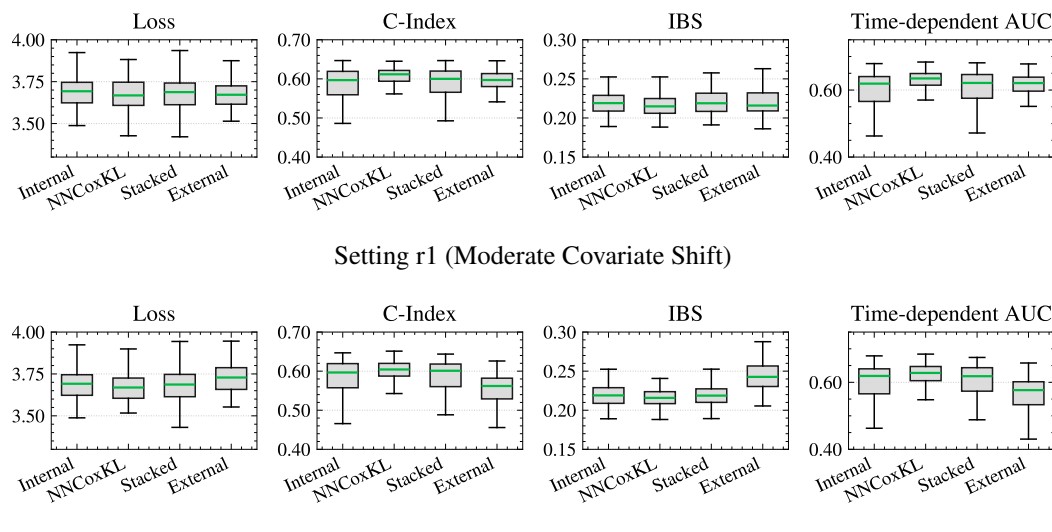

Setting r1 (Moderate Covariate Shift)

Setting r2 (Severe Covariate Shift)

Figure A.2: Comparison of NNCoxKL boxplots under two experimental settings.

sets using a 5:75:20 ratio. The C-index and loss values, calculated on the testing set, served as our evaluation metrics. We compared the performance of models trained using internal data only, and models trained using the external information. The external information came either from a CoxNN model, a linear Cox model fit to the external data, or a CoxNN model fit to a reduced version of the external data where in the external data certain covariates were removed. Specifically, 7 covariates were removed from SUPPORT, 5 from METABRIC, and 4 from GBSG in this heterogeneous setting.

As shown in Table A.2 for all three datasets using a neural network Cox model (NNCoxKL) integrating the best external model (Homo-NN) was better than not using the external information, and was almost as good as a neural network Cox model fit to the large external dataset. For two of the three datasets using the NNCoxKL, integrated with an inferior external model, was better than not using the external information.

A.9    SENSITIVITY ANALYSIS: ROBUSTNESS TO EXTERNAL SCORE SCALING AND TRANSFORMATIONS

In this section, we empirically investigate the robustness of the proposed NNCoxKL framework regarding the scale and distribution of external risk scores. As discussed in Remark 2, we incorporate a temperature-scaling parameter $\alpha$ within the generalized KL divergence term, formulating the external Plackett–Luce probabilities as $\mathbf{p}_k^{(\alpha)}(i) \propto \exp\{\alpha \tilde{r}(\mathbf{Z}_i)\}$.

We assessed the model's stability on the METABRIC dataset, partitioned into internal ($10\%, n = 190$), external ($70\%, n = 1333$), and held-out testing ($20\%, n = 381$) subsets. To ensure a rigorous evaluation, we utilized the Optuna framework to conduct a joint hyperparameter search for the network architecture, integration weight $\eta$, and temperature $\alpha$. Each configuration was optimized over 200 trials. The search space was defined as follows: temperature $\alpha \in [0.1, 10.0]$, integration weight $\eta \in [0.0, 11.0]$, hidden layer dimensions $\in [64, 256]$, number of layers $\in \{1, 2, 3\}$, learning rate $\in [10^{-5}, 10^{-2}]$ (log-uniform), weight decay $\in [10^{-6}, 10^{-2}]$, batch size $\in [32, 128]$, and dropout rate $\in [0.0, 0.5]$.

Performance was evaluated under four distinct transformations of the external risk scores $\tilde{r}$: (1) Original (raw $\tilde{r}$, where $\tilde{r}$ represents the raw risk predictions (linear predictors) derived from a standard Cox Proportional Hazards model fitted to the external dataset;); (2) Shift ($\tilde{r} + 2$) to assess shift invariance; (3) Scale ($0.2 \times \tilde{r}$) to test sensitivity to score compression; and (4) Exponential ($\exp(\tilde{r})$) to evaluate robustness to non-linear monotonic mappings. We report discrimination and

Table A.2: Prediction performance of NNCoxKL and NNCox on the METABRIC, SUPPORT and GBSG datasets. Data were randomly split into internal, external, and testing sets using a 5:75:20 ratio. Performance metrics were averaged over 20 different splits. NNCoxKL = model by the proposed transfer learning method. The external model is either a neural network using all covariates (Homo-NN), a linear Cox model using all covariates (Heter-Cox Linear) or a neural network model using a reduced set of covariates (Heter-NN)

| Data | Internal Model | External Model | C-index (sd) | Loss (sd) |
|---|---|---|---|---|
| METABRIC | NNCoxKL | Homo-NN | 0.643 (0.019) | 2.958 (0.124) |
| | | Heter-Cox Linear | 0.635 (0.021) | 2.971 (0.132) |
| | | Heter-NN | 0.620 (0.021) | 3.009 (0.122) |
| | NNCox | — | 0.613 (0.017) | 3.014 (0.164) |
| | — | NNCox | 0.648 (0.014) | 2.948 (0.124) |
| SUPPORT | NNCoxKL | Homo-NN | 0.615 (0.010) | 4.644 (0.029) |
| | | Heter-Cox Linear | 0.597 (0.011) | 4.652 (0.032) |
| | | Heter-NN | 0.595 (0.014) | 4.665 (0.030) |
| | NNCox | — | 0.591 (0.005) | 4.673 (0.035) |
| | — | NNCox | 0.616 (0.005) | 4.671 (0.033) |
| GBSG | NNCoxKL | Homo-NN | 0.670 (0.015) | 3.112 (0.146) |
| | | Heter-Cox Linear | 0.648 (0.016) | 3.189 (0.119) |
| | | Heter-NN | 0.664 (0.021) | 3.169 (0.138) |
| | NNCox | — | 0.638 (0.013) | 3.158 (0.145) |
| | — | NNCox | 0.674 (0.008) | 3.106 (0.147) |

calibration performance using the Concordance Index (C-Index), Integrated Brier Score (IBS), and Time-Dependent AUC (TD-AUC).

Table A.3 summarizes the performance on the held-out test set. The "Local" baseline denotes an NNCox model trained exclusively on the internal subset. The results demonstrate that NNCoxKL maintains consistent performance improvements over the local baseline across all transformations, validating the efficacy of the joint optimization of $\eta$ and $\alpha$ in adapting to diverse external score distributions.

Table A.3: Sensitivity analysis on the METABRIC dataset (Test Set, $n = 381$). Comparison of NNCoxKL performance under various transformations of the external risk score. All variants utilize joint optimization of $\eta$ and $\alpha$.

| Metric | Local (Internal) | Scale ($\times 0.2$) | Exponential | Shift ($+2$) | Original |
|---|---|---|---|---|---|
| Loss | 2.986 | 2.978 | 2.966 | 2.944 | 2.943 |
| C-Index | 0.595 | 0.612 | 0.634 | 0.633 | 0.634 |
| IBS | 0.211 | 0.208 | 0.203 | 0.199 | 0.199 |
| TD-AUC | 0.611 | 0.632 | 0.661 | 0.659 | 0.660 |

## A.10 MINI-BATCH OPTIMIZATION

From an optimization perspective, we train NNCoxKL with stochastic mini-batches, following Cox-Time (Kvamme et al., 2019). Although the generalized KL-regularized Cox objective is defined on full-data risk sets, mini-batch updates provide a Monte Carlo approximation to the gradients, improving computational scalability and adding mild stochastic regularization. In section A.10.1 below, full-batch and mini-batch training achieve essentially identical predictive performance on METABRIC, indicating that mini-batch optimization is a practical default for our framework.

### A.10.1 IMPACT OF BATCH SIZE: FULL-BATCH VS. MINI-BATCH OPTIMIZATION

A theoretical concern in neural survival modeling is the estimation of the risk set. The Cox partial likelihood ideally requires evaluation over the entire risk set at each event time. Mini-batch training, while computationally efficient, introduces an approximation bias by restricting the risk set to the current batch.

To evaluate the impact of this approximation in our target domain (small-to-moderate sample sizes), we compared the performance of NNCoxKL using Full-Batch versus Mini-Batch. We utilized the METABRIC dataset with the same experimental setup described in Appendix Section A.9. The only difference is the batch size. We used

The results are presented in Table A.10.1. We observed that the performance differences between full-batch and mini-batch optimization are small. The Full-Batch model achieved a slightly lower Loss (2.942 vs. 2.943), while the Mini-Batch model achieved a marginally higher C-index (0.634 vs. 0.633). This suggests that for the small-sized datasets, the approximation bias introduced by mini-batching is minimal, and the stochastic noise may offer slight regularization benefits. Consequently, our framework is robust to the choice of optimization strategy.

| Optimization Strategy | Loss | C-Index | IBS |
|---|---|---|---|
| Full-Batch | 2.942 | 0.633 | 0.199 |
| Mini-Batch | 2.943 | 0.634 | 0.199 |

Table A.4: Performance comparison between Full-Batch and Mini-Batch optimization on the METABRIC dataset.

### A.11 TRANSFER KNOWLEDGE FROM AN UNINFORMATIVE EXTERNAL MODEL (NULL MODEL)

To evaluate the robustness of the proposed NNCoxKL framework against extremely low-quality or non-informative external signals, we conducted a study integrating a "null" external model. We utilized the METABRIC data followed the same settings as described in Appendix A.9. This null external setting was generated by randomly shuffling the correspondence between the covariate vectors and the survival outcomes, thereby breaking the relationship between the covariates and the time-to-event outcome.

Table A.5 summarizes the performance metrics on the independent testing set. As expected, the External (Null) model yields a C-index of 0.485, indicating a predictive capability almost equivalent to random guessing. Despite the lack of signal in the external source, NNCoxKL achieved better performance than the model trained on internal data alone across all metrics (lower Loss and IBS, higher C-index).

This result demonstrates the robustness of the NNCoxKL framework. The slight improvement suggests that the transfer learning provides a beneficial regularization effect, reducing overfitting even when the external model is pure noise.

Table A.5: Performance comparison when transferring knowledge from a Null (randomized) external model. External: The uninformative null model. Internal Only: NNCox trained on internal data. NNCoxKL: The proposed framework trained on the null external model. Lower values are better for Loss and IBS; higher values are better for C-index.

| Metric | External (Null) | Internal Only | NNCoxKL |
|---|---|---|---|
| Loss | 3.012 | 2.986 | 2.977 |
| C-index | 0.485 | 0.595 | 0.602 |
| IBS | 0.219 | 0.211 | 0.209 |

## A.12 APPENDIX E: EFFICIENT HYPERPARAMETER TUNING VIA BAYESIAN OPTIMIZATION

This section presents a comparison of the computational efficiency and predictive performance of the NNCoxKL framework when using standard Grid Search Cross-Validation versus Bayesian optimization (Optuna). To evaluate the potential for more efficient hyperparameter selection, we employed a Tree-structured Parzen Estimator (TPE) to search the hyperparameter space.

This experiment utilizes Simulation Setting r1 (as detailed in Appendix A.7), which introduces a covariate shift between the internal and external populations. We specifically focused on optimizing the transfer learning parameter $\eta$. We compared a standard Grid Search (evaluating $\eta$ at fixed intervals in $[0, 10]$) against Optuna with varying trial budgets. The results, averaged over 100 random replications, are summarized in Table A.6.

Table A.6: Comparison of tuning strategies for the integration parameter $\eta$ under Simulation Setting r1. Results represent Mean (Standard Deviation) over 100 data replicates.

| Method | C-index | Loss ↓ | IBS | Time (s) |
|---|---|---|---|---|
| Optuna (10 Trials) | 0.6070 (0.0171) | 3.6702 (0.0776) | 0.2131 (0.0098) | 10.74 (3.40) |
| Optuna (20 Trials) | 0.6057 (0.0180) | 3.6763 (0.0769) | 0.2147 (0.0107) | 21.84 (6.16) |
| Grid Search (CV) | 0.6046 (0.0186) | 3.6809 (0.0777) | 0.2157 (0.0112) | 23.20 (9.25) |

As presented in Table A.6, Optuna proves to be a highly efficient alternative to standard Grid Search. Optuna with a budget of 10 trials achieved predictive performance that was fully competitive with the more exhaustive 20-point Grid Search. Importantly, this performance was attained with lower computational cost. Additionally, the Bayesian approach demonstrated robust stability across replications with smaller sd. These results confirm that the TPE sampler can effectively identify optimal hyperparameters for NNCoxKL without requiring a dense and computationally expensive grid search.

## A.13 COMPUTATIONAL COMPLEXITY AND RUNTIME ANALYSIS

In this section, we conducted a runtime analysis using the METABRIC dataset to evaluate the computational burden. Experiments were performed on a computing node equipped with *3.0 GHz Intel Xeon Gold 6154 processor*. We varied the internal sample size from $N = 100$ to $N = 1000$. For each sample size, we trained the NNCoxKL model for 200 epochs using full-batch training. To ensure a consistent evaluation of computational cost, we utilized fixed hyperparameters across all trials (2 hidden layers, 128 hidden units, learning rate $1 \times 10^{-3}$, and weight decay $1 \times 10^{-4}$). The process was replicated 100 times to obtain robust estimates.

The results are summarized Figure A.3. The training time scales linearly with the sample size. For a sample size of $N = 1000$, the total training time for 200 epochs averages 1.88 seconds. These results illustrate that for the small-sized scenarios where NNCoxKL is intended to be used, the computational cost is negligible.

## A.14 SUPPORT DATASET

The Study to Understand Prognoses and Preferences for Outcomes and Risks of Treatments (SUPPORT) is a publicly available dataset that is frequently used as a benchmark in survival analysis research. It comprises data from approximately 8,800 hospitalized patients, including demographic information, comorbidities, physiological measurements, and survival outcomes. This data set has been used in various studies to develop and evaluate survival models, including deep learning approaches (Katzman et al., 2018; Lee et al., 2018; Gensheimer & Narasimhan, 2019; Kvamme et al., 2019).

In our study, we used the preprocessed version of the SUPPORT dataset by Katzman et al. (2018). To assess the effectiveness of our proposed NNCoxKL framework in a data-limited setting, we randomly partitioned the SUPPORT dataset into internal and external data sets, consisting of 5% and 95% of the data, respectively. We further split the internal data set into training sets (80%) and

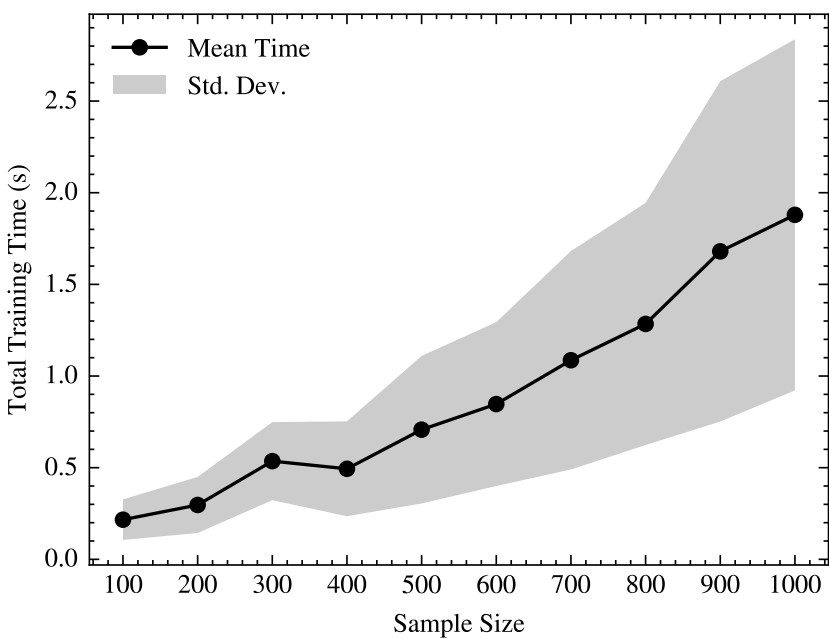

Figure A.3: Training time vs. Sample Size.

testing sets (20%) for model development and evaluation. This partitioning strategy gave a scenario in which the internal dataset is relatively small, reflecting real-world challenges in data availability.

In our analysis of the SUPPORT dataset, the NNCoxKL method, which transfers knowledge from external information, consistently outperformed the NNCox model, which uses only internal data. The NNCox model, trained exclusively on a small subset (5%) of the dataset, showed signs of overfitting as the training progressed. This is evident in Panel A of Figure A.4, where the test loss for NNCox initially declines but then begins to rise, indicating the model's increasing specialization to the training data and its diminishing ability to generalize to new data. In contrast, the NNCoxKL model, leveraging a substantial external data set (95% of SUPPORT) during training, did not show overfitting. Its test loss remained stable or gradually decreased over increasing epochs, suggesting that the model continued to learn generalizable patterns from the expanded dataset.

Panel B of Figure A.4 further illustrates the impact of transfer learning on model complexity. The NNCoxKL model, enriched with external information, demonstrates improved performance with increasing model complexity, reaching optimal performance with three hidden layers. This suggests that transfer learning allows the model to utilize effectively the increased capacity of a more complex architecture to capture intricate relationships in the data. Conversely, the NNCox model without transfer learning consistently shows a higher loss as the number of hidden layers increases, indicating that the model struggles to learn meaningful patterns from the limited internal data and instead overfits to noise as complexity increases. These findings underscore the importance of transfer learning not only in mitigating overfitting but also in enabling the use of more complex model architectures to achieve superior performance, particularly when dealing with limited sample sizes.

## A.15    STAR-CAP SYSTEM

The STAR-CAP system was developed from a study that included 19,684 patients from various centers throughout the United States, Canada, and Europe. The primary goal was to create a system that could more accurately predict prostate cancer-specific mortality (PCSM) than the existing American Joint Committee on Cancer (AJCC) 8th edition staging system. The researchers used a points-based system, assigning points to the following clinical factors: age, T category (tumor size and extent), N category (lymph node involvement), Gleason grade (tumor aggressiveness), pre-treatment PSA levels, and the percentage of positive biopsy cores. These points were then summed to categorize

Figure A.4: This figure evaluates the loss on a separate test set from the SUPPORT dataset for the NNCoxKL model (with transfer learning, solid line) and the NNCox model (without transfer learning, dashed line). Panel A: Loss over training epochs, demonstrating that NNCox exhibits overfitting as epochs increase, while NNCoxKL doesn't suffer from overfitting. Panel B: Loss across varying numbers of hidden layers, illustrating that NNCoxKL initially benefits from increased model complexity, reaching its lowest loss with three hidden layers before slightly increasing with four. In contrast, NNCox demonstrates a consistent upward trend in loss as the number of hidden layers increases. These results highlight the benefits of transfer learning in reducing overfitting issues and facilitating the use of more complicated model architectures for improved performance in survival analysis.

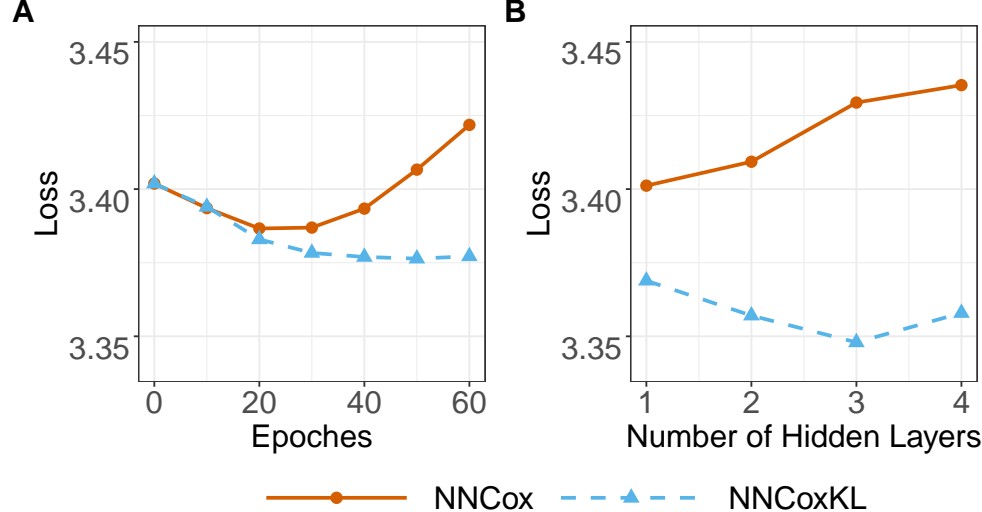

patients into nine distinct risk groups (new stages IA-IIIC). The STAR-CAP scores were originally derived using hazard models where the outcome was prostate cancer specific survival, but with some adaptations, including coarsening of variables and rounding of coefficient estimates to the nearest integer. Thus the final grouping has the property that a higher group has meaningfully worse prognosis than a lower group, but each group is not associated with a specific probability or risk.

The STAR-CAP system was validated using both internal and external datasets, demonstrating superior discriminatory ability and overall performance compared to the AJCC 8th edition system. It also outperformed other commonly used risk stratification systems like the National Comprehensive Cancer Network (NCCN) and the Cancer of the Prostate Risk Assessment (CAPRA).

