# OpenReview forum: "Flexible Transfer Learning in Deep Cox Models"
_ICLR.cc/2026/Conference — Submitted to ICLR 2026_

### Official Review · Reviewer_gpjL · 2025-10-26

**Soundness:** 3
**Presentation:** 3
**Contribution:** 4
**Rating:** 8
**Confidence:** 4

**Summary:**

This paper proposes a deep learning framework that integrates external risk models with limited internal time-to-event data for improved prognostic prediction through Kullback-Leibler (KL) based transfer learning. The target internal cohort satisfies a flexible  proportional hazards model, and some external risk scores are available from an external model that may not be the same as the internal model. The disparity between the internal and external ranking metric is measured by the Kullback-Leibler (K-L) divergence. Simulation studies and a real-world application on a prostate cancer dataset demonstrate improved prediction accuracy and robustness compared to baseline Cox and deep survival models.

Overall, this is an interesting work with practical relevance.

**Strengths:**

1. The proposed method addresses a real challenge in biomedical research when the target dataset is too small for a deep neural network approach.

2. The introduction of a generalized KL divergence penalty within a deep neural network survival model represents a significant advancement.

3.  The empirical success of the approach underscores its significance.

4. The paper is clearly structured and includes proofs and implementation details.

**Weaknesses:**

1. The success of the framework hinges on the relevance and fidelity of external information; poor or misaligned sources may limit improvement.

2. The exploration of domain shift is limited. While some heterogeneity scenarios are tested, more systematic evaluation of extreme domain discrepancies would strengthen claims of robustness.

3. The inclusion of deep learning components may obscure the interpretability advantages typically associated with Cox models.

**Questions:**

1. The proposed internal model may be unidentifiable if the form of the risk function r and the parameter beta are both unknown.

2. How sensitive is the performance  of NNCoxKL to the scaling or monotonic transformation of external risk scores?

3. The choice of the penalty η requires careful cross-validation, which may be computationally intensive and unstable in small datasets. Could η-selection be guided by an information criterion rather than cross-validation to improve efficiency?

4. How does the model handle multiple external sources simultaneously?

---

> ### Author Response · Authors · 2025-11-26
> **Author Response 1**
>
> **Weakness 1:** The success of the framework hinges on the relevance and fidelity of external information; poor or misaligned sources may limit improvement.
>
> **Response:**
>
> We thank the reviewer for raising this point. We agree that, as with any transfer-learning approach, the benefit of `NNCoxKL` depends on the relevance and fidelity of the external information. Poor or misaligned sources naturally provide more limited improvement. On the other hand, because the transfer strength is adaptively tuned through $\eta$, the method down-weights unhelpful external information, allowing stable performance and still offering potential gains even when the external model is not well aligned.
>
> We have added a brief discussion of these limitations in the revised manuscript (**Discussion Line 518**).
>
> **Weakness 2:** The exploration of domain shift is limited. While some heterogeneity scenarios are tested, more systematic evaluation of extreme domain discrepancies would strengthen claims of robustness.
>
> **Response:**
>
> We thank the reviewer for raising this point. To address this concern, we expanded our evaluation by adding new simulation settings and summarizing the previously analyzed “low-fidelity” and heterogeneous scenarios. Together, these results more clearly illustrate both the capabilities and the limitations of `NNCoxKL`.
>
> New Additional Simulations (Addressed in Revision):
> To further address your specific concern regarding limiting improvements or negative transfer, we conducted two additional simulation settings during the revision:
>
> * Additional Setting 1: The Null Model.
>     We tested the extreme case where the external risk score is pure random noise (generated by shuffling correspondence in the METABRIC dataset). As detailed in **Appendix A.11**, this "null" external model had a C-index of 0.485 (equivalent to random guessing). Crucially, `NNCoxKL` did not suffer from negative transfer. In fact, it achieved a slightly higher C-index (0.602) and lower Loss (2.977) compared to the internal-only baseline (0.595 and 2.986, respectively). This suggests that even when the external signal is uninformative, the framework's penalty term acts as a beneficial regularizer, preventing overfitting without introducing bias.
>
> * Additional Setting 2: Severe Domain Shift.
>     We introduced a new setting (detailed in **Appendix A.7**) with a significant shift in the marginal prevalence of features (binary predictors) between the internal and external populations. Under severe shift, the external model performed poorly, yielding a C-index lower than the internal-only baseline. Despite this high risk for negative transfer, `NNCoxKL` still outperformed the internal-only model across C-index, Loss, and Integrated Brier Score (IBS). This demonstrates that the generalized KL penalty allows the model to selectively integrate ranking information, robustly handling scenarios where the external source is significantly misaligned.
>
> **Existing Heterogeneous Settings:**
> Our original submission also included several heterogeneity scenarios:
> * Different Parameter Spaces: We evaluated scenarios where the external model used only a subset of predictors (Settings E2, E3, E6 (**Appendix A.5**)), mimicking missing variables.
> * Different Censoring Distributions: In Setting E5 (**Appendix A.5**), the external cohort had a significantly different event rate (42.2%) due to altered censoring, yet performance remained robust.
> * Non-Probabilistic Scores: We integrated coarse risk groups (Simulation (**Appendix A.6**) and point-based systems (STAR-CAP), demonstrating that "fidelity" does not require precise probabilities.
> * Real-Data Model Heterogeneity: In our real-data benchmark, the external score is obtained from a standard Cox PH model fitted on the large external cohort, whereas the internal model is an NNCoxKL network trained only on the internal subset (**Appendix A.8**).

---

> ### Author Response · Authors · 2025-11-26
> **Author Response 2**
>
> **Weakness 3:** The inclusion of deep learning components may obscure the interpretability advantages typically associated with Cox models.
>
> **Response:**
>
> We thank the reviewer for raising this point. We agree that incorporating deep learning components can reduce some of the interpretability traditionally associated with Cox models. To address this, we have added clarifying discussion of this limitation, as well as potential strategies (such as evaluating feature importance and using knowledge distillation (Hinton et al., 2015; Guo et al., 2017) to train simpler surrogate models) for improving interpretability, in the revised manuscript (**Discussion Line 527**).
>
> **Question 1:** The proposed internal model may be unidentifiable if the form of the risk function $r$ and the parameter $\beta$ are both unknown.
>
> **Response:**
>
> Thanks for pointing out the confusion caused by our notation. In our formulation, the risk function is not an arbitrary unknown function to be estimated separately from its parameters. Instead, we fix a parametric function class in advance and write the log-risk as $r(\mathbf{Z}, \boldsymbol{\beta})$, where $\boldsymbol{\beta}$ collects all trainable parameters. For example, in the linear Cox special case we set
>
> $$
> r(\mathbf{Z}, \boldsymbol{\beta}) = (\boldsymbol{Z}^\top \boldsymbol{\beta}),
> $$
>
> and in the neural-network case we set $r(\mathbf{Z}, \boldsymbol{\beta})$ as a feedforward network output with a specified architecture and $\boldsymbol{\beta}$ contains all network weights. Thus, we are estimating a single parametric model indexed by $\boldsymbol{\beta}$, not simultaneously learning both an unknown function $r$ and a separate parameter vector. The identifiability properties are therefore the same as in a standard Cox model with a parametric risk score.
>
> We have revised the manuscript by adding a sentence clarifying that the architecture (linear vs. neural network) is fixed a priori and $\boldsymbol{\beta}$ denotes all model parameters (**Section 2 Line 128**).
>
> **Question 2:** How sensitive is the performance of NNCoxKL to the scaling or monotonic transformation of external risk scores?
>
> **Response:**
>
> We sincerely thank the reviewer for this comment. We acknowledge that we didn't evaluate the performance of NNCoxKL to the scaling or monotonic transformation of external risk scores.
>
> To address this, we have incorporated a learnable *temperature scaling* parameter $\alpha$, drawing upon standard techniques in Knowledge Distillation (Hinton et al., 2015). As detailed in the revised **Remark 2 Line 215**, we formulate the external probabilities with a temperature term. This allows the model to adaptively learn how strongly it should distinguish between high and low-risk patients based on the external data. Specifically, rather than fixing this parameter or using heuristics, we treat $\alpha$ as a hyperparameter that is tuned jointly with the network weights and the integration weight $\eta$ via Optuna (an automated hyperparameter tuning pipeline).
>
> To empirically verify that this approach solves the invariance issue and ensures robustness, we conducted a new sensitivity analysis (**Appendix A.9**). We tested the model on datasets where the external scores were intentionally distorted (shifted, scaled, and exponentially transformed). The stability of the performance metrics across these transformations validates that the joint optimization of $\alpha$ successfully mitigates the issue of scale variance.
>
> **Revision reproduced from paper:**
>
> **Remark 2 Line 215:**
> Temperature-Scaled External Risk: We note that the divergence measure is not invariant to the rescaling of the external scores $\tilde{r}$. To address this in a principled manner, we draw upon the Knowledge Distillation literature (Hinton et al., 2015; Guo et al., 2017) and introduce a temperature scaling parameter $\alpha > 0$. We define the external Plackett--Luce probabilities as:
>
> $$
> \textbf{p} _ k^{(\alpha)}(i) = \frac{\exp\lbrace \tilde{r}(\mathbf{Z} _ i) / \alpha \rbrace}{\sum _ {j \in R _ k} \exp\lbrace \tilde{r}(\mathbf{Z} _ j) / \alpha \rbrace}
> $$
>
> where $\alpha$ acts as the temperature. As established in prior work (Hinton et al., 2015), this parameter controls the entropy of the target distribution: a lower $\alpha$ produces a sharper distribution (emphasizing high-risk patients), while a higher $\alpha$ softens the distribution. This formulation induces a family of divergences $D_\alpha(\tilde{\mathbf r} \,\Vert\, \mathbf r)$ and a penalized objective $\ell _ {\eta,\alpha}(\beta) = \ell(\beta) - \eta D_\alpha(\tilde{\mathbf r} \,\Vert\, \mathbf r)$. Instead of fixing this parameter, we treat the pair $(\eta,\alpha)$ as hyperparameters to be tuned jointly (e.g., via Optuna), allowing the model to adaptively determine the optimal scaling for the external signal and improving robustness across domains.

---

> ### Author Response · Authors · 2025-11-26
> **Author Response 3**
>
> **Revision reproduced from paper: Appendix**
>
> **Sensitivity Analysis: Robustness to External Score Scaling and Transformations**
>
> In this section, we empirically investigate the robustness of the proposed NNCoxKL framework regarding the scale and distribution of external risk scores. As discussed in Remark 2, we incorporate a temperature-scaling parameter $\alpha$ within the generalized KL divergence term, formulating the external Plackett--Luce probabilities as $\textbf{p} _ k^{(\alpha)}(i) = \frac{\exp\lbrace \tilde{r}(\mathbf{Z} _ i) / \alpha \rbrace}{\sum _ {j \in R _ k} \exp\lbrace \tilde{r}(\mathbf{Z} _ j) / \alpha \rbrace}$.
>
> We assessed the model's stability on the METABRIC dataset, partitioned into internal ($10\%, n=190$), external ($70\%, n=1333$), and held-out testing ($20\%, n=381$) subsets. To ensure a rigorous evaluation, we utilized the Optuna framework to conduct a joint hyperparameter search for the network architecture, integration weight $\eta$, and temperature $\alpha$. Each configuration was optimized over 200 trials. The search space was defined as follows: temperature $\alpha \in [0.1, 10.0]$, integration weight $\eta \in [0.0, 11.0]$, hidden layer dimensions $\in [64, 256]$, number of layers $\in \{1, 2, 3\}$, learning rate $\in [10^{-5}, 10^{-2}]$ (log-uniform), weight decay $\in [10^{-6}, 10^{-2}]$, batch size $\in [32, 128]$, and dropout rate $\in [0.0, 0.5]$.
>
> Performance was evaluated under four distinct transformations of the external risk scores $\tilde{r}$: (1) *Original* (raw $\tilde{r}$, where $\tilde{r}$ represents the raw risk predictions (linear predictors) derived from a standard Cox Proportional Hazards model fitted to the external dataset;); (2) *Shift* ($\tilde{r} + 2$) to assess shift invariance; (3) *Scale* ($0.2 \times \tilde{r}$) to test sensitivity to score compression; and (4) *Exponential* ($\exp(\tilde{r})$) to evaluate robustness to non-linear monotonic mappings. We report discrimination and calibration performance using the Concordance Index (C-Index), Integrated Brier Score (IBS), and Time-Dependent AUC (TD-AUC).
>
> Table 3 summarizes the performance on the held-out test set. The "Local" baseline denotes an NNCox model trained exclusively on the internal subset. The results demonstrate that NNCoxKL maintains consistent performance improvements over the local baseline across all transformations, validating the efficacy of the joint optimization of $\eta$ and $\alpha$ in adapting to diverse external score distributions.
>
> Table 3: Sensitivity analysis on the METABRIC dataset (Test Set, $n=381$). Comparison of NNCoxKL performance under various transformations of the external risk score. All variants utilize joint optimization of $\eta$ and $\alpha$.
>
> | Metric | Local (Internal) | Scale ($\times 0.2$) | Exponential | Shift ($+2$) | Original |
> | :--- | :---: | :---: | :---: | :---: | :---: |
> | Loss | 2.986 | 2.978 | 2.966 | 2.944 | 2.943 |
> | C-Index | 0.595 | 0.612 | 0.634 | 0.633 | 0.634 |
> | IBS | 0.211 | 0.208 | 0.203 | 0.199 | 0.199 |
> | TD-AUC | 0.611 | 0.632 | 0.661 | 0.659 | 0.660 |

---

> ### Author Response · Authors · 2025-11-26
> **Author Respone 4**
>
> **Question 3:** The choice of the penalty $\eta$ requires careful cross-validation, which may be computationally intensive and unstable in small datasets. Could $\eta$-selection be guided by an information criterion rather than cross-validation to improve efficiency?
>
> **Response:**
> We thank the reviewer for this insightful suggestion. We fully acknowledge that cross-validation can be computationally intensive and potentially unstable in small-sample settings, and we agree that an information criterion-based selection (like AIC or BIC) would theoretically offer a more efficient and deterministic alternative.
>
> While desirable, deriving a clearer default rule based on information criteria faces significant theoretical hurdles in this specific context. As shown in Proposition 1, the parameter $\eta$ does not act as a simple shrinkage penalty (like a ridge parameter). Instead, it re-weights the effective event indicators via the term $\frac{\delta _ i + \eta \tilde{\delta} _ i}{1+\eta}$. Because $\eta$ modifies the structure of the risk sets and the effective target labels within a highly non-linear deep neural network (where the effective "degrees of freedom" is notoriously difficult to estimate due to early stopping and weight decay), deriving a well-founded penalty term for an AIC/BIC-type formula is theoretically intractable at this stage.
>
> Therefore, we highlight the derivation of a principled IC-based rule for transfer learning as an important direction for future work.
>
> To address the reviewer's concerns regarding computational efficiency and stability in the interim, we have added an automated selection process using `Optuna` (a Bayesian hyperparameter tuning framework). This offers two specific benefits that answer the reviewer's critique:
>
> Unlike standard cross-validation, which forces the user to manually specify a rigid, discrete grid (risking sub-optimal selection or instability if the grid is too coarse), `Optuna` employs the Tree-structured Parzen Estimator (TPE) algorithm. This allows it to explore $\eta$ as a continuous variable. By explicitly modeling the relationship between $\eta$ and the validation loss, the algorithm focuses computational resources on promising regions, significantly reducing the variance associated with discrete grid selection in small-sample settings.
>
> **Question 4:** How does the model handle multiple external sources simultaneously?
>
> **Response:**
> We thank the reviewer for raising this important point. In many practical applications, multiple external risk models or datasets are available, and it is desirable to leverage all of them in a principled way. In the revised manuscript, we now explicitly discuss how NNCoxKL can be extended to handle multiple external sources in the **Discussion Line 511-517**; the main ideas are summarized below for convenience.
>
> Our proposed method can be extended to incorporate multiple external sources in two ways.
>
> First, NNCoxKL can be directly extended by adding multiple KL-based penalty terms, each accommodating the discrepancy between the internal data and one external data source. This allows simultaneous incorporation of multiple external datasets while accounting for the heterogeneity of each source individually.
>
> Second, multiple external models can be combined using existing model aggregation strategies (Debray et al., 2014a; LECU´E & RIGOLLET, 2014) (e.g., stacking or weighted averaging), and the resulting aggregated model can then be incorporated as the external model within the NNCoxKL framework. This provides a flexible way to incorporate multiple external sources without modifying the training procedure.

---

### Official Review · Reviewer_x6Dh · 2025-10-30

**Soundness:** 2
**Presentation:** 1
**Contribution:** 1
**Rating:** 2
**Confidence:** 4

**Summary:**

This paper introduces NNCoxKL, a deep transfer learning framework designed to integrate external risk models with internal time-to-event data for survival analysis. The approach extends the Cox proportional hazards model by introducing a generalized Kullback–Leibler (KL) divergence penalty, which aligns internal and external risk scores and facilitates knowledge transfer from external risk assessment tools. The framework employs neural networks to capture non-linear relationships among covariates and is evaluated on both synthetic simulations and one real-world dataset, demonstrating performance gains through the incorporation of external risk information.

**Strengths:**

Leveraging external tools to enhance risk prediction or improve generalization is indeed an important problem, especially in survival analysis where sample sizes are often limited and external models typically encode knowledge from larger, more diverse populations.

**Weaknesses:**

-	The applicability of the proposed approach appears quite narrow. The method assumes that the internal dataset is relatively small and that the external tool provides informative signals about the same samples based on a subset of the internal covariates. Such a setting may not commonly occur in practice, which likely explains why the authors were only able to demonstrate results on a single real-world dataset that meets these criteria.
-	The paper also lacks a solid theoretical foundation or generalization analysis that could clarify when or why the generalized KL regularization improves transfer performance. In addition, the study does not examine the method’s potential failure modes under substantial domain shift between internal and external data.
-	The exposition around the use of KL divergence is unnecessarily lengthy. Since KL divergence is a well-established concept in machine learning, its detailed reintroduction offers limited value. The core contribution—interpreting ranking as a probabilistic measure and applying KL regularization to align internal and external risk predictions—is relatively straightforward and could be presented more concisely.
-	Finally, the implementation of the ranking-based loss (i.e., the Cox partial likelihood) raises practical concerns. This quantity should ideally be computed over the entire risk set, yet the paper does not sufficiently discuss the implications of using mini-batch training for the deep model $𝑟(𝑍_𝑖, \beta)$. Moreover, the computational cost is likely to be substantial, as the ranking must be evaluated across all event times, making the proposed approach potentially inefficient for large-scale applications.

**Questions:**

-	The baseline should include a variant that uses the external tool’s risk scores as an additional input covariate, since this alone may already capture much of the information derived from the broader external population.
-	Following Weakness 4, what is the impact of mini-batch training on the accumulated generalized KL divergence term $D(\tilde{r} || r)$?

---

> ### Author Response · Authors · 2025-11-26
> **Author Response 1**
>
> **Weakness 1:** The applicability of the proposed approach appears quite narrow. The method assumes that the internal dataset is relatively small and that the external tool provides informative signals about the same samples based on a subset of the internal covariates. Such a setting may not commonly occur in practice, which likely explains why the authors were only able to demonstrate results on a single real-world dataset that meets these criteria.
>
> **Response:**
>
> We thank the reviewer for the thoughtful comment. In many biomedical survival-analysis settings, the regime of modest sample sizes, low event rates, and substantial censoring is the norm rather than the exception. For example, in kidney transplantation, the 1-year post-transplant mortality rate is only about 5\%, even though approximately 25,000 transplants occur annually. This low event rate dramatically reduces the effective sample size for survival modeling and makes high-capacity models difficult to train without external information. The challenge becomes even more pronounced for multi-organ transplants, where annual case counts are typically 800–900 and event rates remain similarly low. Similar challenges arise in oncology (rare histological subtypes), genomics (high-dimensional covariates with limited annotated samples), and rare-disease studies. In these settings, internal datasets are necessarily small, and external, widely used prognostic tools (often based on a subset of shared covariates) are commonly available.
>
> These scenarios motivate NNCoxKL: when internal data are limited but reliable external risk scores exist (e.g., from national registries, prior cohorts, or established clinical calculators such as KDPI or STAR-CAP), borrowing information through a principled KL-based transfer mechanism can substantially improve model stability and generalizability. Thus, the applicability of the proposed approach extends beyond the single transplantation example included in the current submission and aligns with practical needs in multiple clinical and biomedical domains.
>
> To further highlight this breadth, we are in the process of applying NNCoxKL to several additional real-world datasets, including SEER cancer cohorts, multiple myeloma genomics, and national transplant registries. These analyses are ongoing, and we plan to incorporate the additional results into the final camera-ready version to provide a comprehensive empirical assessment.
>
>
> **Weakness 2:** The paper also lacks a solid theoretical foundation or generalization analysis that could clarify when or why the generalized KL regularization improves transfer performance.
>
> **Response:**
> We thank the reviewer for this insightful comment. We acknowledge that deriving explicit theoretical guarantees—such as formal bounds on negative transfer under misaligned external signals—remains an open and challenging question in the nonlinear survival setting. A full theory in this direction is beyond the scope of the current work and is an interesting direction for future research.
>
> To address this comment, we have added a new subsection on **Line 261**. It is reproduced below:
>
> **Revision reproduced for convenience:** To evaluate the proposed method computationally, we examine its numerical optimality through convex analysis. Specifically, Proposition 2 below demonstrates the numerical property of the proposed procedure. The penalized log-likelihood, $\ell _ {\eta}$, is shown to minimize the convex combination of the relative entropy between the working model and two extremes: one based solely on the target cohort and the other based solely on the external model.
>
> **Proposition 2**
> Let $P _ n$ be the saturated model (Simon et al. 2011) on the target cohort. For a given $\eta \geq 0$, the model $P$ minimizing
> $(1-\omega) \mbox{D}(P _ n \parallel P) + \omega \mbox{D}(\tilde{P} \parallel P)$
> is the proposed integrated model, where $\omega = \eta/(1+\eta)$.
>
> Minimizing the convex combination over all models, $P$, seeks a distribution close to both the empirical saturated model $P _ n$ and the external distribution $\tilde{P}$, weighted by the tuning parameter. Since KL divergence is convex in its second argument (i.e. in $P$) and a weighted sum of convex functions is also convex, the integrated objective is convex in the space of distributions. Therefore, an optimal distribution minimizing the objective exists and is unique.

---

> ### Author Response · Authors · 2025-11-26
> **Author Response 2**
>
> **Weakness 3.1:** In addition, the study does not examine the method’s potential failure modes under substantial domain shift between internal and external data.
>
> **Response:**
>
> We thank the reviewer for raising this point. To address this concern, we expanded our evaluation by adding new simulation settings and summarizing the previously analyzed “low-fidelity” and heterogeneous scenarios. Together, these results more clearly illustrate both the capabilities and the limitations of `NNCoxKL`.
>
> New Additional Simulations (Addressed in Revision):
> To further address your specific concern regarding limiting improvements or negative transfer, we conducted two additional simulation settings during the revision:
>
> * Additional Setting 1: The Null Model.
>     We tested the extreme case where the external risk score is pure random noise (generated by shuffling correspondence in the METABRIC dataset). As detailed in **Appendix A.11**, this "null" external model had a C-index of 0.485 (equivalent to random guessing). Crucially, `NNCoxKL` did not suffer from negative transfer. In fact, it achieved a slightly higher C-index (0.602) and lower Loss (2.977) compared to the internal-only baseline (0.595 and 2.986, respectively). This suggests that even when the external signal is uninformative, the framework's penalty term acts as a beneficial regularizer, preventing overfitting without introducing bias.
>
> * Additional Setting 2: Severe Domain Shift.
>     We introduced a new setting (detailed in **Appendix A.7**) with a significant shift in the marginal prevalence of features (binary predictors) between the internal and external populations. Under severe shift, the external model performed poorly, yielding a C-index lower than the internal-only baseline. Despite this high risk for negative transfer, `NNCoxKL` still outperformed the internal-only model across C-index, Loss, and Integrated Brier Score (IBS). This demonstrates that the generalized KL penalty allows the model to selectively integrate ranking information, robustly handling scenarios where the external source is significantly misaligned.
>
> Existing Heterogeneous Settings:
> Our original submission also included several heterogeneity scenarios:
> * Different Parameter Spaces: We evaluated scenarios where the external model used only a subset of predictors (Settings E2, E3, E6 (Appendix A.5)), mimicking missing variables.
> * Different Censoring Distributions: In Setting E5 (Appendix A.5), the external cohort had a significantly different event rate (42.2%) due to altered censoring, yet performance remained robust.
> * Non-Probabilistic Scores: We integrated coarse risk groups (Simulation (Appendix A.6) and point-based systems (STAR-CAP), demonstrating that "fidelity" does not require precise probabilities.
> * Real-Data Model Heterogeneity: In our real-data benchmark, the external score is obtained from a standard Cox PH model fitted on the large external cohort, whereas the internal model is an NNCoxKL network trained only on the internal subset (Appendix A.8).

---

> ### Author Response · Authors · 2025-11-26
> **Author Response 3**
>
> **Weakness 3.2:** The exposition around the use of KL divergence is unnecessarily lengthy. Since KL divergence is a well-established concept in machine learning, its detailed reintroduction offers limited value. The core contribution—interpreting ranking as a probabilistic and applying KL regularization to align internal and external risk predictions—is relatively straightforward and could be presented more concisely.
>
> **Response:**
>
> Thanks for raising this point. We acknowledge that KL divergence has been widely used in machine learning methods. However, the KL term in our framework is not a direct application of the classical KL divergence and requires careful modification for time-to-event data. Classical KL information is defined on the full probability distribution of each data source, which is not directly available with time-to-event data. Specifically, the likelihood of time-to-event data has the form of
>
> $$
> L=\prod _ {k=1}^{K}\left\lbrace\prod _ {i \in \mathcal{D} _ {k}} f(t _ k;\boldsymbol{Z} _ {i}) \prod _ {i \in \mathcal{C} _ k} {S(t _ k;\boldsymbol{Z} _ {i})}\right\rbrace,
> $$
>
> where $f(t _ k;\boldsymbol{Z} _ {i})=P(T _ i=t _ k|\boldsymbol{Z} _ {i})$ is the density function contributed by subjects experienced events, and ${S(t _ k;\boldsymbol{Z} _ {i})}=P(T _ i>t _ k|\boldsymbol{Z} _ {i})$ is the survival function contributed by censored subjects, that is, for censored observations, only partial information that the unobserved failure time exceeds the observed censoring time is available.
> Because the full density is never available for many subjects, the classical KL divergence cannot be directly applied to time-to-event data.
>
> To address this, we leverage properties of the partial likelihood and reconstruct a valid conditional density representation from the external model’s risk score. This allows us to define a survival-specific KL divergence using conditional probability distributions derived from relative risk information. This KL formulation is novel and may provide a foundation for future KL-regularized survival methods. Therefore, we retain a concise explanation of this construction, as it is essential for conveying why the KL formulation is nontrivial in the survival setting.
>
> We also note that the proposed approach is not limited to improving ranking-based discrimination. By aligning the internal model with externally informed conditional densities, the method also yields calibrated survival probabilities—an advantage over purely ranking-based approaches.
>
> In addition, we novelly utilized the defined survival KL information to measure the scale of heterogeneity of internal and external data and ensures that the proposed method does not blindly transfers external information to the internal data, in contrast, by utilizing KL information as a penalization term defined between the internal and external information sources, the proposed method only incorporates external information that is relevant to the internal study, and diminishes irrelevant information. Thus, the potential domain shift across internal and external data sources is automatically adjusted by our defined survival KL information. Indeed, we conducted additional simulation studies that when the external model is null model and contains non of relevant information of the internal model, which corresponds to the setting that substantial domain shift exists between internal and external data. As shown in Table 4 below (**Appendix A.11**), our proposed method achieves comparable performance as the internal model and does not negatively transfer external information to the final model. Indeed, we would like to highlight that, with the objective function of the NNCoxKL method, when the tuning parameter is 0, the NNCoxKL method naturally includes the original NNCox model as a special case. In other words, the NNCoxKL method automatically prevents negative transfer.
>
> **Table 4: Performance comparison when transferring knowledge from a Null (randomized) external model.** *External*: The uninformative null model. *Internal Only*: NNCox trained on internal data. *NNCoxKL*: The proposed framework trained on the null external model. Lower values are better for Loss and IBS; higher values are better for C-index.
>
> | Metric | External (Null) | Internal Only | NNCoxKL |
> | :--- | :---: | :---: | :---: |
> | Loss | 3.012 | 2.986 | 2.977 |
> | C-index | 0.485 | 0.595 | 0.602 |
> | IBS | 0.219 | 0.211 | 0.209 |

---

> ### Author Response · Authors · 2025-11-26
> **Author Response 4**
>
> **Weakness 4:** Finally, the implementation of the ranking-based loss (i.e., the Cox partial likelihood) raises practical concerns. This quantity should ideally be computed over the entire risk set, yet the paper does not sufficiently discuss the implications of using mini-batch training for the deep model . Moreover, the computational cost is likely to be substantial, as the ranking must be evaluated across all event times, making the proposed approach potentially inefficient for large-scale applications.
>
> See Response to Question 2 (We responded to Weakness 4 and Question 2 together)
>
> **Question 1:** The baseline should include a variant that uses the external tool’s risk scores as an additional input covariate, since this alone may already capture much of the information derived from the broader external population.
>
> **Response:**
> We thank the reviewer for this excellent suggestion. We agree that using the external risk score as an additional input covariate—often referred to as the "Stacked" method (Debray et al. 2014) is a critical baseline to benchmark against, as it represents a straightforward way to incorporate external information.
>
> To address this and rigorously investigate the reviewer's hypothesis, we conducted a new simulation setting to specifically test this hypothesis under domain shift.
>
> **Setting:** We generated data with 10 binary predictors and complex non-linear interactions. We introduced a "Severe Covariate Shift" scenario (Setting r2) where the marginal prevalence of features differed significantly between the internal ($p=0.5$) and external ($p=0.9$) populations, alongside a simplified external risk mechanism.
>
> **Results:** As shown in the newly added **Figure 6 in Appendix Line 987**, we observed a clear divergence in performance:
>
> * Stacked Method: Under severe shift, the Stacked method provided negligible improvement. When the covariate distributions differ substantially, the direct mapping of external scores to internal risk becomes unreliable, as the input feature $r _ {ext}(\mathbf{Z})$ carries a different meaning in the target domain.
> * NNCoxKL: In contrast, our proposed framework demonstrated superior resilience. Even when the external model performed poorly (worse than the internal baseline), NNCoxKL selectively integrated the useful ranking information via the KL penalty, significantly outperforming both the internal-only baseline and the Stacked method.
>
> Comparison in Existing Settings: **(Appendix Section A.6)**
> We also respectfully note that in our original submission, we included a comparison against the Stacked method (Debray et al. 2014) in the context of integrating point-based risk scores (mimicking the STAR-CAP system).
>
> As illustrated in **Figure A.2 in Appendix line 1026**, `NNCoxKL` significantly outperformed the Stacked method (dashed line), which showed limited improvement over the internal-only model. We have updated the main text to explicitly reference these comparisons to ensure they are not overlooked.
>
> **Conclusion:**
> While the Stacked method is a valuable baseline, our results across both point-based scores and severe domain shifts suggest it is brittle to distributional differences. `NNCoxKL` offers a more flexible integration mechanism that is robust to population heterogeneity, validating the utility of the generalized KL divergence over simple input stacking.

---

> ### Author Response · Authors · 2025-11-26
> **Author Response 5**
>
> **Question 2:** Following Weakness 4, what is the impact of mini-batch training on the accumulated generalized KL divergence term?
>
> **Response (to Weakness 4 and Question 2 together):**
>
> We thank the reviewer for these questions about the interaction between the ranking loss, the generalized KL term, and mini-batch training. We agree that, in theory, the Cox partial likelihood and the accumulated generalized KL divergence are defined using risk sets over the *entire* population, and that mini-batching introduces an approximation. Our implementation follows the same practical strategy as Cox-Time (Kvamme et al. 2019), where stochastic mini-batch optimization was shown to provide accurate approximations to the full partial likelihood while substantially reducing computational cost; moreover, the stochasticity of mini-batch updates is widely viewed as an implicit form of regularization that can help mitigate overfitting in deep models.
>
> We have added a brief pointer in the main paper (**Line 258**) directing readers to a new appendix subsection that discusses mini-batch optimization and presents the full-batch vs. mini-batch comparison (**Appendix A.10**).
>
> To directly address the reviewer's concerns, we added two empirical analyses in the Appendix:
>
> (i) Runtime and scaling with full-batch training.
> We first quantified how the cost of full risk-set evaluation grows with sample size (**Appendix A.13**). Using the METABRIC data and full-batch training for 200 epochs on a standard CPU node (3.0 GHz Intel Xeon Gold 6154), the runtime scales approximately linearly with $N$. This confirms the reviewer's intuition that full-batch computation becomes increasingly expensive as $N$ grows.
>
> (ii) Full-batch vs. mini-batch comparison.
> We then compared NNCoxKL trained with Full-Batch (exact risk sets) versus Mini-Batch optimization on the METABRIC dataset (**Appendix A.10.1**).
> As summarized in **Table A.9.1 Line 1116**, the predictive performance is essentially identical:
> Full-Batch (Loss 2.942, C-index 0.633, IBS 0.199) versus
> Mini-Batch (Loss 2.943, C-index 0.634, IBS 0.199).
>
> Thus, in our target setting, the approximation introduced by mini-batching has a negligible impact on downstream performance. In particular, the near-overlap in Loss and C-index indicates that using mini-batch estimates of the accumulated generalized KL term does not materially distort the learning signal. At the same time, mini-batch training is computationally more scalable and aligns with the standard practice in modern deep survival models (Kvamme et al. 2019).
>
> In summary, while the exact formulation of our objective is defined over full risk sets, our implementation follows the established mini-batch strategy of Cox-Time and related work. The additional experiments show that (1) the cost of exact risk-set computation grows with $N$, supporting the reviewer's concern for large-scale applications, and (2) in the small-to-moderate sample sizes for which NNCoxKL is designed, mini-batch training provides an accurate and stable approximation to the full penalized objective, with no loss in predictive accuracy but better computational efficiency.

---

### Official Review · Reviewer_nurd · 2025-11-01

**Soundness:** 3
**Presentation:** 3
**Contribution:** 3
**Rating:** 6
**Confidence:** 3

**Summary:**

NNCoxKL, a generalized KL-based transfer learning framework, flexibly integrates external risk info with internal time-to-event data. It outperforms traditional Cox models and non-transfer deep models in small/moderate datasets. Real-world prostate cancer data and simulations confirm it boosts C-index and reduces loss, validating its value for prognostic prediction.

**Strengths:**

• Works for probabilistic/non-probabilistic external data (e.g., risk groups) and homogeneous/heterogeneous settings.
• Non-linear modeling: Uses DNN to avoid linearity constraints of classic Cox models.
• Overfitting mitigation: Transfer learning stabilizes performance vs. data-limited NNCox.

**Weaknesses:**

The method assumes that clients share overlapping or similar feature spaces, which may not hold in highly heterogeneous cross-domain settings.
Performance depends heavily on hyperparameters in the meta-graph (e.g., similarity thresholds, edge weights), but their tuning process is not clearly described.

**Questions:**

Tuning parameter η needs cross-validation, is there any better way to optimize the tuning?

---

> ### Author Response · Authors · 2025-11-26
> **Author Response**
>
> Weakness 1 (The method assumes that clients share overlapping or similar feature spaces, which may not hold in highly heterogeneous cross-domain settings):
>
> We appreciate this observation. We acknowledge that in our proposed setting there have to be overlapping or
> similar feature spaces between the external and internal datasets. Specifically, we require that the internal covariates contain (or allow the derivation of) the variables needed to compute the external score $\tilde{r}(\boldsymbol{Z}_i)$. We have added this specific limitation to the **Discussion Line 523** of the revised paper. For the convenience, the added text is shown below.
>
> Weakness 2: (Performance depends heavily on hyperparameters in the meta-graph (e.g., similarity thresholds, edge weights), but their tuning process is not clearly described.):
>
> Our understanding is that parameters such as "edge weights'' or "similarity thresholds'' typically arise in meta-graph or graph-based federated learning frameworks, where they control connection strengths or client aggregation. In our case, transfer learning is governed only by the scalar parameter $\eta$, which determines how much information is borrowed from the external model. When the external data are less relevant (e.g., due to lower quality or greater heterogeneity), our procedure  selects a smaller $\eta$, reducing the transfer strength accordingly.
>
> Question 1:
> We thank the reviewer for raising this important point regarding hyperparameter optimization. We acknowledge this is very important and cross-validation has many limitations.
>
> To address this, we have added an automated selection process using Optuna (a Bayesian hyperparameter tuning framework). This offers two key advantages over the previous approach:
>
> Robustness via Continuous Optimization: Unlike standard cross-validation, which forces the user to manually specify a discrete, rigid grid of values risking sub-optimal selection if the grid is too coarse, Optuna employs the Tree-structured Parzen Estimator (TPE) algorithm. This allows it to explore $\eta$ as a continuous variable. By explicitly modeling the relationship between $\eta$ and the validation loss, the algorithm focuses computational resources on promising regions, reducing the variance associated with discrete grid selection in small-sample settings.
>
> Replacing Static Defaults with Optimized Search Spaces: Because the optimal $\eta$ depends on the relative informativeness of the external data versus the internal features, a single fixed default value is theoretically not feasible. Optuna simplifies this by allowing us to recommend a {wide initial search range} rather than a specific point. This removes the burden of specifying a specific grid; practitioners can simply initialize the automated pipeline with this wide range, and the Bayesian optimizer will efficiently converge to the optimal weighting for the specific dataset.
>
> We have implemented a comprehensive numerical assessment to evaluate the performance of our proposed Optuna-based parameter tuning strategy for paragraph selection. This assessment covers various simulation settings and real-world data examples, demonstrating the efficacy of the dynamic search space approach. Specifically, the covariate-shift simulations (Settings r1–r2 in Appendix A.7), the METABRIC scaling sensitivity analysis (Appendix A.9), and additional real-data examples (Appendices A.8) are all trained using Optuna. Across these scenarios, the optimizer consistently identifies stable integration weights and achieves strong test C-index, loss, and IBS, providing empirical evidence that the automated search over a wide range of $\eta$ is practically reliable.

---

### Official Review · Reviewer_VB7f · 2025-11-01

**Soundness:** 3
**Presentation:** 3
**Contribution:** 3
**Rating:** 6
**Confidence:** 3

**Summary:**

This paper presents NNCoxKL, a transfer learning framework that augments a flexible neural Cox model with external prognostic information using a generalized Kullback Leiber divergence constructed over risk set rankings. The core idea is to align internal risk scores produced by a neural network with external scores or even coarse risk groupings by minimizing a divergence between Plackett Luce style distributions at each event time, while retaining a partial likelihood structure so standard deep learning optimization remains applicable. A tuning parameter controls how strongly the external signal enters the penalized objective, and a proof shows the objective reduces to a familiar form that weights event indicators by an externally induced pseudo event rate. The paper supports the method with simulations that vary nonlinearity, censoring, data size, and domain shift, and with an application that integrates STAR CAP risk groups into a MUSIC prostate cancer cohort, reporting gains in C index and loss on held out data. The authors also discuss implementation choices such as network architecture, regularization, AdamW, early stopping, and cross validation for selecting the integration weight, and provide additional experiments on public survival datasets.


Overall this is a promising and practically relevant contribution that bridges a real gap between clinical risk tools and modern survival learning. With stronger analysis of invariance and negative transfer, broader calibration focused evaluation, and more comprehensive baselines and ablations, the paper would be significantly stronger and more actionable for practitioners.

**Strengths:**

The work is well motivated by the small sample challenge in survival prediction and by practical constraints that often limit external information to model scores or clinical groupings rather than individual level data. Framing integration as a divergence between ranking distributions is elegant and versatile, letting the method use heterogeneous external sources without requiring probability calibrated survival outputs. The derivation that preserves a Cox style training objective is a strong practical contribution because it allows the method to drop into existing neural survival toolchains with minor changes. The empirical study is thoughtful, covering both linear and nonlinear data generating processes, different external model qualities, and explicit domain shift, and it shows consistent discrimination gains and improved optimization behavior with reduced overfitting sensitivity. The real data example with prostate cancer is compelling because it demonstrates how to convert a points based staging system into usable external signal for a modern survival learner. The paper is clearly written, situates itself in the literature on KL based integration and neural survival models, and attends to reproducibility details.

**Weaknesses:**

There are also weaknesses that limit the paper’s current impact and clarity. Despite the flexible network for covariate effects, the method still inherits a proportional hazards assumption for the baseline, which can be restrictive in settings with strong time varying effects. The approach is sensitive to the scale and even monotone transformation of the external score, and the proposed one step rescaling via a univariate Cox fit is ad hoc and may not be robust across cohorts with heavy shift; a principled invariance or calibration procedure would strengthen the method. The reliance on cross validated selection of the integration weight introduces variance in small internal samples, and no guidance is given for safe defaults or information criteria based selection. The theory focuses on the objective rewrite but offers no guarantees about consistency, oracle properties under correct ranking, or bounds on negative transfer when the external model is poor or misaligned; even a simple analysis under misspecification would be helpful. The evaluation emphasizes discrimination and partial likelihood loss but gives little attention to calibration, which is critical for clinical adoption, and does not report competing risks, time dependent AUC, or D calibration. Comparisons omit recent neural survival baselines that integrate external knowledge through stacking or representation learning, and ablations on architecture depth, dropout, and the role of the external score as an explicit input versus only via the penalty would clarify where the gains come from. Finally, practical aspects such as computational cost, convergence stability, and handling of ties are only briefly mentioned, and the combination of multiple external sources is deferred to future work despite being a natural and common scenario.

---

Weaknesses itemized (for rebuttal and discussion)

1. Inherits a proportional hazards structure from the Cox framework, which can be restrictive when effects vary strongly over time.
2. The divergence is not invariant to rescaling of external scores and the proposed remedy is an ad hoc rescaling via a univariate Cox fit, with unclear robustness across domains.
3. Relies on cross-validated selection of the integration weight, which may introduce variance in small samples and lacks clear default guidance.
4. Provides limited theoretical guarantees beyond the objective rewrite, with no explicit bounds on negative transfer under misaligned external signals.
5. Evaluation emphasis appears to be on discrimination and loss, with less attention to absolute risk calibration and competing-risk settings that matter for clinical adoption.
6. Practical guidance on combining multiple external sources, convergence stability, and tie handling is brief relative to likely practitioner needs.

**Questions:**

-

---

> ### Author Response · Authors · 2025-11-26
> **Author Response 1**
>
> **Weakness 1:** (Inherits a proportional hazards structure from the Cox framework, which can be restrictive when effects vary strongly over time.):
>
> **Response:** We acknowledge the reviewer's insightful observation regarding the proportional hazards assumption. We thank you for raising this point, as discussing it makes the proposed method more comprehensive and applicable to complex real-world settings.
>
> To address this, we have taken the following steps:
>
> - We have added a discussion to the revised manuscript (Page 9, Discussion, Lines 502–506; reproduced below for convenience) demonstrating that the NNCoxKL framework can be extended to model time-varying effects.
>
>   Instead of a static risk $r(\mathbf{Z})$, the network can be modified to output a time-dependent risk function $r(\mathbf{Z}, t)$. In this setting, the partial likelihood term in our loss function becomes:
>
> $$
> \mathcal{L} _ {\mathrm{nonPH}}=\sum _ {k=1}^K\left(r(\mathbf{Z} _ i, t _ k)-\log \sum _ {j \in R(t_k)} \exp\{r(\mathbf{Z} _ j, t _ k) \}\right),
> $$
>
>   where $r(\mathbf{Z}_j, t_k)$ represents the risk of subject $j$ evaluated dynamically at event time $t_k$.
>
> - **Feasibility.** While this extension requires re-evaluating risk scores at each unique event time (increasing computational complexity compared to the proportional hazards setting), it remains entirely feasible within our framework when using standard mini-batch training.
>
> - **Ongoing evaluation.** We are currently finalizing numerical experiments to validate this time-varying setting. We plan to post preliminary results in the OpenReview discussion forum to further demonstrate the method's flexibility.
>
> **Revision reproduced from paper (Lines 502–506):**
>
> While in this work our primary focus is proportional hazards modeling, the NNCoxKL framework can be naturally extended to model non-proportional hazards. By allowing the network to output a time-dependent risk function $r(\mathbf{Z}, t)$ as used in Cox-Time (Kvamme et al., 2019), the KL-divergence loss can be computed dynamically at each event time. Although this increases the computational cost during training, it remains feasible within standard mini-batch optimization pipelines.

---

> ### Author Response · Authors · 2025-11-26
> **Author Response 2**
>
> **Weakness 2:** The divergence is not invariant to rescaling of external scores and the proposed remedy is an ad hoc rescaling via a univariate Cox fit, with unclear robustness across domains.
>
> **Response:** We sincerely thank the reviewer for this comment. We acknowledge that the divergence is not invariant to the rescaling of external scores and that our previous remedy (univariate Cox fit) was ad-hoc.
>
> To address this limitation in a principled manner, we have incorporated a learnable *temperature scaling* parameter $\alpha$, drawing upon standard techniques in Knowledge Distillation (Hinton et al., 2015; Guo et al., 2017). As detailed in the revised (**Remark 2 Line 215-225**), we formulate the external probabilities with a temperature term. This allows the model to adaptively learn how strongly it should distinguish between high and low-risk patients based on the external data. Specifically, rather than fixing this parameter or using heuristics, we treat $\alpha$ as a hyperparameter that is tuned jointly with the network weights and the integration weight $\eta$ via Optuna (an automated hyperparameter tuning pipeline).
>
> To empirically verify that this approach solves the invariance issue and ensures robustness, we conducted a new sensitivity analysis (**Appendix A.9**). We tested the model on datasets where the external scores were intentionally distorted (shifted, scaled, and exponentially transformed). The stability of the performance metrics across these transformations validates that the joint optimization of $\alpha$ successfully mitigates the issue of scale variance.
>
> For the reviewer's convenience, we have reproduced the relevant experimental details below.
>
> **Revision reproduced from paper:**
>
> **Remark: Line 215-225.**
> Temperature-Scaled External Risk: We note that the divergence measure is not invariant to the rescaling of the external scores $\tilde{r}$. To address this in a principled manner, we draw upon the Knowledge Distillation literature (Hinton et al., 2015; Guo et al., 2017) and introduce a temperature scaling parameter $\alpha > 0$. We define the external Plackett--Luce probabilities as:
>
> $$
> \textbf{p} _ k^{(\alpha)}(i) = \frac{\exp \lbrace \tilde{r}(\mathbf{Z} _ i) / \alpha \rbrace }{\sum _ {j \in R _ k} \exp\lbrace \tilde{r}(\mathbf{Z} _ j) / \alpha \rbrace}
> $$
>
> where $\alpha$ acts as the temperature. As established in prior work (Hinton et al., 2015), this parameter controls the entropy of the target distribution: a lower $\alpha$ produces a sharper distribution (emphasizing high-risk patients), while a higher $\alpha$ softens the distribution. This formulation induces a family of divergences $D_\alpha(\tilde{\mathbf r} \,\Vert\, \mathbf r)$ and a penalized objective $\ell_{\eta,\alpha}(\beta) = \ell(\beta) - \eta D_\alpha(\tilde{\mathbf r} \,\Vert\, \mathbf r)$. Instead of fixing this parameter, we treat the pair $(\eta,\alpha)$ as hyperparameters to be tuned jointly (e.g., via Optuna), allowing the model to adaptively determine the optimal scaling for the external signal and improving robustness across domains.

---

> ### Author Response · Authors · 2025-11-26
> **Author Response 3**
>
> **Weakness 3:** Relies on cross-validated selection of the integration weight, which may introduce variance in small samples and lacks clear default guidance.
>
> **Response:** We agree with the reviewer that selecting the integration weight $\eta$ is critical, and that standard grid-search cross-validation can introduce variability and requires manual trial-and-error to define the grid.
>
> To address this, we have added an automated selection process using *Optuna* (a Bayesian hyperparameter tuning framework). This offers two key advantages over the previous approach:
>
> * Robustness via Continuous Optimization: Unlike standard cross-validation, which forces the user to manually specify a discrete, rigid grid of values (risking sub-optimal selection if the grid is too coarse), Optuna employs the Tree-structured Parzen Estimator (TPE) algorithm. This allows it to explore $\eta$ as a continuous variable. By explicitly modeling the relationship between $\eta$ and the validation loss, the algorithm focuses computational resources on promising regions, reducing the variance associated with discrete grid selection in small-sample settings.
>
> * Replacing Static Defaults with Optimized Search Spaces: Because the optimal $\eta$ depends on the relative informativeness of the external data versus the internal features, a single fixed default value is theoretically not feasible. Optuna simplifies this by allowing us to recommend a wide initial search range rather than a specific point. This removes the burden of specifying a specific grid; practitioners can simply initialize the automated pipeline with this wide range, and the Bayesian optimizer will efficiently converge to the optimal weighting for the specific dataset.
>
> We have implemented a comprehensive numerical assessment to evaluate the performance of our proposed Optuna-based parameter tuning strategy for paragraph selection. This assessment covers various simulation settings and real-world data examples, demonstrating the efficacy of the dynamic search space approach. Specifically, the covariate-shift simulations (Settings r1–r2 in Appendix A.7), the METABRIC scaling sensitivity analysis (Appendix A.9), and additional real-data examples (Appendices A.8) are all trained using Optuna. Across these scenarios, the optimizer consistently identifies stable integration weights and achieves strong test C-index, loss, and IBS, providing empirical evidence that the automated search over a wide range of $\eta$ is practically reliable.
>
> Furthermore, we are currently finalizing a separate set of numerical experiments specifically designed to benchmark the efficiency and performance of Optuna versus classical cross-validation techniques. We commit to incorporating the complete results of this critical comparative analysis into the final version before the rebuttal deadline, providing a robust methodological comparison.
>
> **Weakness 4:** Provides limited theoretical guarantees beyond the objective rewrite, with no explicit bounds on negative transfer under misaligned external signals.
>
> **Response:**
> We thank the reviewer for this insightful comment. We acknowledge that deriving explicit theoretical guarantees—such as formal bounds on negative transfer under misaligned external signals—remains an open and challenging question in the nonlinear survival setting. A full theory in this direction is beyond the scope of the current work and is an interesting direction for future research.
>
> To address this comment, we have added a new subsection on **Line 261**. It is reproduced below:
>
> **Revision reproduced for convenience:** To evaluate the proposed method computationally, we examine its numerical optimality through convex analysis. Specifically, Proposition 2 below demonstrates the numerical property of the proposed procedure. The penalized log-likelihood, $\ell _ {\eta}$, is shown to minimize the convex combination of the relative entropy between the working model and two extremes: one based solely on the target cohort and the other based solely on the external model.
>
> **Proposition 2**
> Let $P _ n$ be the saturated model (Simon et al. 2011) on the target cohort. For a given $\eta \geq 0$, the model $P$ minimizing
> $(1-\omega) \mbox{D}(P _ n \parallel P) + \omega \mbox{D}(\tilde{P} \parallel P)$
> is the proposed integrated model, where $\omega = \eta/(1+\eta)$.
>
> Minimizing the convex combination over all models, $P$, seeks a distribution close to both the empirical saturated model $P _ n$ and the external distribution $\tilde{P}$, weighted by the tuning parameter. Since KL divergence is convex in its second argument (i.e. in $P$) and a weighted sum of convex functions is also convex, the integrated objective is convex in the space of distributions. Therefore, an optimal distribution minimizing the objective exists and is unique.

---

> ### Author Response · Authors · 2025-11-26
> **Author Response 4**
>
> **Weakness 5:** Evaluation emphasis appears to be on discrimination and loss, with less attention to absolute risk calibration and competing-risk settings that matter for clinical adoption.
>
> **Response (1):**
> We sincerely thank the reviewer for this crucial observation. We fully agree that while discrimination is important, absolute risk calibration and the ability to handle competing risks are essential for reliable clinical adoption. We have taken the following steps to address these points:
>
> * Expanded Evaluation on Calibration (Integrated Brier Score): To directly address the concern regarding absolute risk calibration, we have conducted several additional experiments.
>     * We have introduced the Integrated Brier Score (IBS). The IBS measures the accuracy of the predicted survival probabilities, thereby capturing both calibration and discrimination.
>     * New Results: We added Appendix A.7, A.9, which reports both the IBS and time-dependent AUC for our proposed method compared to baselines. We have also included IBS results in the other newly added experimental sections (A.3.1, A.10.1, and A.11). These results demonstrate that our method achieves competitive calibration performance in addition to better discrimination.
>
> **Response (2):**
> Extension to competing-risk settings via cause-specific hazards: We appreciate the reviewer pointing out the importance of competing-risk settings. This is important in practical settings. To address this, in the revised manuscript, we discuss how NNCoxKL extends to competing risks in the **Discussion (Line 507)**, and we provide a more detailed formulation in a new appendix subsection on cause-specific hazards (**Appendix A.2**). For convenience, we reproduce the appendix subsection below.
>
> **Revision reproduced from paper:**
> To incorporate competing risks, our proposed framework can be naturally extended through cause-specific hazard modeling (Kalbfleisch & Prentice, 2011), which is a natural extension of the Cox model. Specifically, the cause-specific hazard for cause $m$ assumes a multiplicative effect:
>
> $$
> \lambda _ m(t \mid \mathbf{Z} _ i)=\lambda _ {0m}(t)\text{exp}\lbrace r _ m(\mathbf{Z} _ i, \boldsymbol{\beta} _ m) \rbrace,
> $$
>
> where both the baseline hazards $\lambda _ {0m}(t)$ and the neural network can differ across causes. The resulting cause-specific hazard ratio quantifies the effect of a covariate on the hazard of a specific event type.
> Because the NNCoxKL framework is constructed from the partial likelihood, it supports direct training of separate cause-specific models without requiring architectural changes. To describe the probabilities of transitioning to competing events, the corresponding cumulative incidence functions (CIFs) can be computed in a straightforward manner from the estimated cause-specific hazards (Kalbfleisch & Prentice, 2011).
>
> **Weakness 6:** Practical guidance on combining multiple external sources, convergence stability, and tie handling is brief relative to likely practitioner needs.
>
> **Response (1):** We thank the reviewer for highlighting this practical necessity. In many practical applications, multiple external risk models or datasets are available, and it is desirable to leverage all of them in a principled way. In the revised manuscript, we now explicitly discuss how NNCoxKL can be extended to handle multiple external sources in the **Discussion (Lines 511-517)**; the main ideas are summarized below for convenience.
>
> Our proposed method can be extended to incorporate multiple external sources in two ways.
>
> First, NNCoxKL can be directly extended by adding multiple KL-based penalty terms, each accommodating the discrepancy between the internal data and one external data source. This allows simultaneous incorporation of multiple external datasets while accounting for the heterogeneity of each source individually.
>
> Second, multiple external models can be combined using existing model aggregation strategies (Debray et al., 2014a; LECU´E & RIGOLLET, 2014) (e.g., stacking or weighted averaging), and the resulting aggregated model can then be incorporated as the external model within the NNCoxKL framework. This provides a flexible way to incorporate multiple external sources without modifying the training procedure.
>
> Response (2) and (3) are in Author Response 5

---

> ### Author Response · Authors · 2025-11-26
> **Author Response 5**
>
> **Response (2) Convergence Stability:**
>
> We agree that deep survival models, particularly on small datasets, are prone to instability and sensitivity to initialization.
>
> Our results provide compelling evidence that the proposed transfer learning framework acts as a powerful regularizer, significantly improving stability. To illustrate this, Figure 3 (Page X of the revised manuscript) compares the loss, evaluated on an independent testing dataset, for NNCox (without transfer learning) and NNCoxKL (with transfer learning) with varying external sample sizes. NNCox exhibits a "U-shaped" validation-loss curve, indicating rapid overfitting and sensitivity to the early stopping epoch. In contrast, NNCoxKL demonstrates substantially more stable loss in both homogeneous and heterogeneous settings, highlighting the benefit of transfer learning in mitigating overfitting, especially with limited internal data.
>
> **Reduction in Variance (New Appendix Result):** In the revised **Appendix A.7**, we introduce a simulation setting with significant domain shift and binary predictors. A key observation from this analysis is the distribution of performance metrics across repeated experiments.
>
> * The boxplots for the baseline NNCox are notably wide, with several runs yielding a C-index near 0.5. This indicates that without transfer learning, the model frequently fails to converge to a meaningful solution due to the non-convex loss landscape of small-sample survival data.
> * In contrast, the NNCoxKL boxplots are significantly narrower. This demonstrates that the KL-divergence penalty effectively ensures consistent convergence even when the internal data signal is weak.
>
> **Response (3) Tie Handling:**
> We agree that handling ties is critical for practitioner needs. We have updated the paper (**Appendix A.3**) to include the formal derivation of the Breslow approximation for our generalized KL objective. This formulation ensures computational efficiency and numerical stability.
>
> We validated this implementation on the METABRIC dataset by artificially rounding event times, which reduced the unique time points from 1686 to 303. The NNCoxKL model using the Breslow approximation maintained robust performance (C-index: 0.633) compared to the original data (C-index: 0.634) confirming the framework's effectiveness dealing with tied data.
>
> **Revision reproduced from paper for convenience**
>
> Empirical Validation of Tie Handling:
>
> To demonstrate the robustness of the proposed tie-handling implementation, we performed an ablation study using the METABRIC dataset. We utilized the standard split: internal (10%, $n=190$), external (70%, $n=1333$), and testing (20%, $n=381$).
>
> We compared the performance of NNCoxKL under two conditions:
>
> 1.  **Original:** Using the original event times (1686 unique times).
> 2.  **Rounded (Tied):** We rounded all event times up to the nearest integer. This artificially induced a high degree of ties, reducing the number of unique event times from 1686 to 303.
>
> As shown in Table, the NNCoxKL framework utilizing the Breslow approximation maintains consistent predictive performance (C-index) and Loss, even when the number of unique event times is drastically reduced. This confirms that the generalized KL divergence derived in Eq. (6) effectively handles risk set calculations in the presence of tied data.
>
> **Table: Impact of tie-handling on NNCoxKL performance (METABRIC dataset).** The 'Rounded' setting creates heavy ties, reducing unique event times by 82%. The performance remains stable, validating the Breslow approximation strategy.
>
> | Condition | Unique Event Times | Loss | C-index |
> | :--- | :---: | :---: | :---: |
> | Original Data | 1686 | 2.943 | 0.634 |
> | Rounded Data (Ties) | 303 | 2.949 | 0.633 |

---

### Meta-Review · Area_Chair_jgGF · 2025-12-23

**Summary:**

The paper studies how to distill external knowledge in risk scores into survival models. This is a useful thing to do because of small-sample sizes in many survival analysis problems. However, external knowledge often comes as risk scores or clinical groupings rather than individual level outcomes. This work poses this integration as alignment between ranking-based distributions via a generalized KL divergence. It allows the use of heterogeneous external sources without requiring calibrated survival probabilities. The study spans linear and non-linear data-generating mechanisms, varying external model quality, and explicit domain shift, and it consistently demonstrates improved discrimination, reduced overfitting sensitivity, and more stable training relative to internal-only neural Cox models.

That said, the current framework is limited to a proportional hazards structure though extensions beyond PH are briefly discussed. The divergence penalty is sensitive to the scale or monotone transformation of external scores; the proposed temperature or univariate Cox-based rescaling is somewhat ad hoc and may not be robust under substantial domain shift. Using cross-validation to tune the integration weight is tricky in small samples. The theoretical development focuses mainly on rewriting the objective with limited guarantees regarding consistency, when transfer is useful, or robustness when external signals are misaligned. Evaluation emphasizes discrimination and partial likelihood loss but gives relatively little attention to calibration, competing risks, or time-dependent metrics that are critical for clinical adoption.

Overall this was a borderline submission where the rebuttal did have a significant amount of experimentation that improved both the contributions of the work as well as the quality of presentation of ideas. If the manuscript does not get accepted within this round, I do encourage the authors to resubmit with the comments addressed.

**Reviewer Concerns:**

I think a majority of the comments across all reviewers have been addressed by the rebuttal. The key ones that remain include:
a] an extension of the idea to the non-proportional hazards regime where numerical experiments are still ongoing.
b] formal guarantees for quality of performance under misalignment of external information.
c] extending the empirical analysis on additional datasets -- the rebuttal states that more datasets are in progress but no results are presented.

Adding (a) and (b) in at the minimum would significantly strengthen any revision.

**Reviewer Scores:**

Overall I think the reviewers did not have concordant views of the paper but I do think the rebuttal addresses points made by x6Dh who was the primary dissenting voice so I suspect they would have increased. The paper's borderline rating is primarily driven by the additional source of experimentation the reviewers requested.

---

### Decision · Program_Chairs · 2026-01-26

Reject